


# Comprehensive Automobile Research System (CARS) – a Python-based Automobile Emissions Inventory Model

Bok H. Baek[1], Rizzieri Pedruzzi[2], Minwoo Park[3], Chi-Tsan Wang[1], Younha Kim[3], Chul-Han Song[4], and Jung-Hun Woo[3]

[1]Center for Spatial Information Science and Systems – George Mason University, Fairfax, VA, USA.

[2]Department of Sanitary and Environmental Engineering, Federal University of Minas Gerais, Belo Horizonte, Brazil.

[3]Department of Advanced Technology Fusion, Konkuk University, Republic of Korea

[4]School of Earth and Environmental Engineering, Gwangju Institute Science and Technology, Republic of Korea

*corresponding to: Jung-Hun Woo (jwoo@konkuk.ac.kr)*

## Abstract

The Comprehensive Automobile Research System (CARS) is an open-source python-based automobile emissions inventory model designed to efficiently estimate high quality emissions from motor-vehicle emission sources. It can estimate the criteria air pollutants, greenhouse gases, and air toxics in various temporal resolutions at the national, state, county, and any spatial resolution based on the spatiotemporal resolutions of input datasets. The CARS is designed to utilize the local vehicle activity database, such as vehicle travel distance, road link-level network Geographic Information System (GIS) information, and vehicle-specific average speed by road type, to generate a temporally and spatially enhanced automobile emissions inventory for policymakers, stakeholders, and the air quality modeling community. The CARS model adopted the European Environment Agency's (EEA) onroad automobile emissions calculation methodologies to estimate the hot exhaust, cold start, and evaporative emissions from onroad automobile sources. It can optionally utilize road link-specific average speed distribution (ASD) inputs to reflect more realistic vehicle speed variations by road type than a road-specific single averaged speed approach. Also, utilizing high-resolution road GIS data allows the CARS to estimate the road link-level emissions to improve the inventory's spatial resolution. When we compared the official 2015 national mobile emissions from Korea's Clean Air Policy Support System (CAPSS) against the ones estimated by the CARS, there is a moderate increase of VOC (33%), CO (52%), and fine particulate matter ($PM_{2.5}$) (15%) emissions while NOx and SOx are reduced by 24% and 17% in the CARS estimates. The main differences are driven by the usage of different vehicle activities and the incorporation of road-specific ASD, which plays a critical role



in hot exhaust emission estimates but wasn't implemented in Korea's CAPSS mobile emissions
inventory. While 52% of vehicles use gasoline fuel and 35% use diesel, gasoline vehicles only
contribute 7.7% of total NOx emissions while diesel vehicles contribute 85.3%. But for VOC
emissions, gasoline vehicles contribute 52.1% while diesel vehicles are limited to 23%. While
diesel buses are only 0.3% of vehicles, each vehicle has the largest contribution to $NO_x$ emissions
(8.51% of $NO_x$ total) due to its longest daily VKT. For VOC, CNG buses are the largest contributor
with 19.5% of total VOC emissions. It indicates that the CNG bus is better for the rural area while
the diesel bus is better applicable for the urban area for a better ozone control strategy because the
rural area is usually $NO_x$ limited for ozone formation and urban area is VOC limited region. For
primary $PM_{2.5}$, more than 98.5% is from diesel vehicles. The CARS model's in-depth analysis
feature can assist government policymakers and stakeholders develop the best emission abatement
strategies.
Keywords: inventory: automobile, vehicle emissions, hot exhaust, cold start, evaporative, python

# 1    Introduction

Globally, ambient pollution causes more than 4.2 million premature deaths every year. Indoor
air pollution causes 3.8 million deaths and over 90% of people live in places where air pollutants
exceed the WHO standards (WHO, 2019; Hogrefe et al., 2001a; Hogrefe et al., 2001b; Dennis et
al., 2010; Rao et al., 2011; Appel et al., 2013; Luo et al., 2019). To effectively mitigate air
pollutants, both developed and developing countries' governments have been implementing
stringent air pollution abatement control policies to reduce harmful regional air pollutants.
Chemical transport models (CTM) are a powerful tool to study and develop an efficient control
strategy for local and regional air quality (Hogrefe et al., 2001a; Hogrefe et al., 2001b; Dennis et
al., 2010; Rao et al., 2011; Appel et al., 2013; Luo et al., 2019). The CTM simulation results
strongly rely on precise input data, such as emission inventory, meteorology, land surface
parameters, and chemical mechanisms in the atmosphere. The most dominant factor for accurate
CTM performance is temporally and spatially high-quality emissions, especially in the worst air
quality regions with significant anthropogenic emission sources.

The major anthropogenic emission sources in urban areas are from transportation emission
sectors. The tailpipe emissions from the vehicle's combustion process contain many air pollutants,
including nitrogen oxides (NOx), volatile organic compounds (VOCs), carbon monoxide (CO),
ammonia ($NH_3$), sulfur dioxide ($SO_2$), and primary particulate matter (PM) which will participate
in the formation of detrimental secondary pollutants like ozone and $PM_{2.5}$ in the atmosphere. In
the Seoul Metropolitan Area (SMA) in South Korea, transportation automobile sources contribute
the most to the total $NO_X$ and primary $PM_{2.5}$ emissions across all emission sources. While more
than 60% of total ambient $PM_{2.5}$ are primary $PM_{2.5}$ directly emitted from the sources, (Choi et al.,





2014; Kim et al., 2017a; Kim et al., 2017b; Kim et al., 2017c), the rest of the ambient $PM_{2.5}$ are
secondary pollutant from heterogenous chemical reactions in the atmosphere during the
transportation. Thus, it is critical to understand and represent better on the emission patterns from
the transportation automobile sources in the CTM model. The use of process-based automobile
emission models is highly recommended to meet the needs in CTM model because it can estimate
the high quality spatiotemporal automobile emissions based on parameterizations of the emission
processes, such as physical, chemical, and vehicle operation processes from on/off-network roads
(Moussiopoulos et al., 2009; Russell and Dennis, 2000).

There are two methodologies known in emission inventory development: top-down and
bottom-up. The choice of methods is determined by the input data availability. The top-down
approach primarily relies on the aggregated and generalized country or regional information,
especially in developing countries where only limited datasets and information are available. It has
its limitations on representing the vehicle emission process realistically due to the lack of detailed
activity and ancillary supporting data. However, the bottom-up approach requires higher-quality
spatiotemporal activity datasets like road network information, vehicle composition (vehicle type,
engine size, vehicle age, and fuel-technology), pollutant-specific emissions factors, road segment
length, traffic activity data, and fuel consumption (EEA, 2019; Ibarra-Espinosa et al., 2018b;
IEMA, 2017). It can generate more accurate and detailed automobile emissions across various
operating processes, such as hot exhaust, evaporative, idling, and hot soak (Nagpure et al., 2016;
Ibarra-Espinosa et al., 2018a).

There are several bottom-up mobile emissions models available, like MOVES (MOtor
Vehicle Emissions Simulator) from the U.S. Environmental Protection Agency (USEPA), the
European Environment Agency's (EEA) model COPERT (COmputer Programmed to calculate
Emissions from Road Transport), the HERMES (High-Elective Resolution Modelling Emission
System) from Barcelona Supercomputing Center (Guevara et al., 2019), the VEIN (Vehicular
Emissions INventory) model developed by Ibarra-Espinosa et al. (2017), and the VAPI (Vehicular
Air Pollution Inventory) model developed by Nagpure and Gurjar (2012) for India (Nagpure et al.,
2016). While these models are all bottom-up emission inventory models, a single model cannot
meet all modelers, policymakers, and stakeholders' needs because each model holds its own pros
and cons. They are developed differently to meet their own needs based on the types of traffic
activity and emission factors, emission calculation methodologies, and other optional/available
traffic-related inputs such as average speed distribution and geographical resolution. The bottom-
up emission calculations can be further complicated when other factors like emissions factors with
varying vehicle operation speeds and local meteorology are accounted for.

The MOVES model has the strength to generate high-quality emissions for up to 16
different emission processes (i.e., Running Exhaust, Start Exhaust, Evaporative, Refueling,
Extended Idling, Brake, Tire, etc.). It can simulate not only county-level but also road segment
level depending on data availability. It can also reflect local meteorological conditions, such as





ambient temperature and relative humidity, which can significantly impact both pollutants and emissions processes (Choi et al., 2017; Perugu et al., 2018). Disadvantages of this model are the lack of transparency for emission factors and algorithms and that it is computationally expensive to generate these high-quality emissions inventories (Li et al., 2016; Xu et al., 2016; Liu et al., 2019; Perugu, 2019). The COPERT model that is widely used in European countries has its advantages, such as the capability to model emissions in high resolution. Additionally, it is fully integrated with the EEA's onroad vehicle emissions factors guidelines and can generate a complete quality assurance (QA) and visualization summary (Ntziachristos et al., 2009). The cons are that it is a proprietary commercial licensed software, limited to EEA guidance, and challenging to modify and update with any key input datasets like the latest emission factors from non-European countries (Lejri et al., 2018; Rey DR, 2018; Li et al., 2019; Lv et al., 2019; Smit et al., 2019).

The HERMES and VEIN are both recently released bottom-up inventory models. They have their pros in that they are both open-source models based on open-source computing languages (Python and R), which provide transparency of emission calculations with a considerable amount of data behind it (Ibarra-Espinosa et al., 2018b; Guevara et al., 2019). Both models are driven by comma-separated value (CSV) formatted input files, making it very easy for users to modify the input datasets. They are also based on the EEA's emission calculation method and equipped with a complete QA and visualization tool based on Python and R libraries. However, it is not an easy task to update the emission factors, and generate other required input datasets for other countries, and lacks support for any control strategy plan feature to generate a responsive reduced emissions inventory for policymakers, stakeholders, and modelers.

The VAPI (Vehicular Air Pollution Inventory) model was developed in India because the country does not have an extensive and robust traffic-related dataset to run these kinds of vehicular emissions inventory models (Nagpure et al., 2016; Perugu, 2019).

There are also a few shortcomings of incorporating these bottom-up models into CTM studies. These models require strong programming skills to operate, such as collecting and preparing the input data to fit the model requirement, configuring the model variables, and changing specific variables that may be hidden somewhere. Another downside is that while the administration-level emissions inventory can be estimated by those models, it requires a 3rd party emissions processor like the SMOKE (Sparse Matrix Operator Kerner Emissions) modeling system (Baek and Seppanen, 2021) to process and generate spatially and temporally resolved emissions inputs for CTM. Some detailed information, like link-level hourly driving patterns, can be lost in the emissions processing steps.

There is no single model capable of meeting all the requirements across various spatial and temporal scales (Pinto et al., 2020). However, transparency, simplicity, and a user-friendly interface are requirements for those who mainly work in transportation policy and air quality modeling development (Fallahshorshani et al., 2012; Kaewunruen et al., 2016; Sallis et al., 2016;


Sun et al., 2016; Tominaga and Stathopoulos, 2016). Thus, the ideal mobile emissions modeling
system would be computationally optimized, easy-to-use, and have a user-friendly interface.
Additionally, the model should easily adapt detailed local activity information and the state-of-art
emission factors as an input to represent them in the highest resolution possible in time and space.
We have developed the Comprehensive Automobile Research System (CARS) to meet these
requirements, especially for the air quality research community, policymakers, and air quality
modelers. The CARS is a stand-alone, fully modularized, computationally optimized, python-
based automobile emission model. The modularization improves the efficiency of processing times.
Once district and road-link level annual/monthly/daily total emissions are computed, the rest of
the processes are optional. It can generate chemically speciated, spatially gridded hourly emissions
for CTMs without any 3$^{rd}$ party emissions modeling system to develop the highest quality CTM-
ready emissions inputs. All functions are operated by independent modules and can be enabled by
users. Details on modularization will be discussed later. The CARS model can be easily adopted
and is simple for users to add new functions or modules in the future. The application of the CARS
to South Korea will be described in detail later.

## 2    CARS Emissions Calculation

The CARS is an open-source Python-based customizable motor vehicle emissions
processor that estimates onroad and offroad emissions for specific criteria and toxic air pollutants.
Figure 1 is a schematic of the CARS overview. It applies vehicle, engine, and fuel specific
emission factors to traffic data to estimate the local level annual, monthly, and daily total emissions
inventory. The emissions inventory calculations require the list of pollutant-specific emissions
factors by vehicle age, local activity data, average speed profile/distribution by road type, and
geographic information system (GIS) road segment shapefiles inputs. The spatial resolution of
VKT defines the CARS geographic scale (i.e. district, county, state, and country) for emission
calculations. Unlike the district-level Korea Clean Air Policy Support System (CAPSS)
automobile emission inventory (Lee et al., 2011a; Lee et al., 2011b), the CARS applies high-
resolution annual average daily traffic (AADT) data from the road GIS shapefiles to distribute the
total district emissions into road link-level emissions. Optionally, these road link-level emissions
can be used to generate spatially gridded CTM-ready emissions input data once the output
modeling domain is defined. How the CARS estimates spatially and temporally enhanced
automobile emissions inventories will be discussed in detail next chapter.
South Korean traffic databases by the Korea CAPSS team (Lee et al., 2011b) from the
National Institute of Environmental Research (NIER) were used in this study to compute the
updated onroad automobile emissions inventory. The databases include individual vehicle activity
data (daily total VKT), road activity data (average speed distribution by road), vehicle age specific
emission factors, road type information, surface weather data, and GIS road shapefiles.





## 2.1    Individual Daily Total VKT Activity Data


The accuracy of vehicle emissions inventories from CARS significantly depends on the
quality of traffic density information. To accurately represent traffic density for the CARS, this
study imported the national registered vehicle-specific daily total VKT from South Korea's
Vehicle Inspection Management System (VIMS), which belongs to the Korea Transportation
Safety Authority (KTSA). It contains over 50 million records from 2013 to 2017. For the CARS
model, we first sorted these records by the vehicle identification number (VIN) to remove any
duplicates and then built vehicle-specific daily total VKT traffic activity data in the CSV format.
The summary of those vehicle numbers and VKTs is presented in Fig. 2. Sedan vehicles using
gasoline fuel comprise the greatest percentage of total vehicles at 47% (~10.4 million) and have
the highest VKT. Most vehicles demonstrate similar patterns between the number of vehicles and
daily VKT. However, as expected, LPG (liquefied petroleum gas)-fueled taxi are high in VKT
compared to the number of vehicles due to their daily long distance travel pattern.
Besides the numbers of vehicles, the vehicle type ($v$) and the VIN are applied to individual
vehicles to calculate their daily total VKT or $VKT_{v,age}$ (km d$^{-1}$). In Eq. (1), the individual vehicle
VKT with the manufactured year ($VKT_{v,age}$) is calculated based on the cumulative mileage ($M_f$)
since the last inspection date ($D_f$) and registration date ($D_0$). Korea's NIER defines the vehicle
types (Ryu et al., 2003; Ryu et al., 2004; Ryu et al., 2005; Lee et al., 2011a) based on a combination
of vehicle types (e.g., sedan, truck, bus, etc), engine sizes (e.g., compact, full size, midsize, etc)
and fuel types (e.g., gasoline, diesel, LPG, etc). Full details of vehicle types and daily total VKT
are shown in Appendix A and B.
$$VKT_{v,age} = \frac{M_{f;v,age}}{D_{f;v,age} - D_{0;\,v,age}} \qquad (1)$$

## 2.2    Emission Calculations


Automobile emission sources cover motorized engine sources from network (onroad) and
off-network (nonroad). Nonroad transportation sources represent any motorized engine vehicle
emissions that occurred from off-network roads, such as aviation, railways, construction, and boats.
Onroad automobile emissions are ones that occur on the network roads. While nonroad automobile
emissions are important, we will focus on the onroad automobile emissions from network roads
using their local traffic-related datasets. The following section explains the approach of the onroad
automobile emission processes. The onroad emission ($E_{onroad}$) in the CARS is defined in Eq. (2),
which includes three major emission processes (Ntziachristos and Samaras, 2000):
$$E_{onroad} = E_{hot} + E_{cold} + E_{vap} \qquad (2)$$





The hot exhaust emissions ($E_{hot}$) are the vehicle's tailpipe emissions when the internal combustion engine (ICE) combusts the fuel to generate energy under the average operating temperature. The cold start emissions ($E_{cold}$) are the tailpipe emissions from the ICE when the cold vehicle engine is ignited and the operational temperature is below average condition. The evaporative VOC emissions ($E_{vap}$) are the emissions evaporated/permeated from the fuel systems (fuel tanks, injection systems, and fuel lines) of vehicles.

The CARS first applies the hot exhaust emission factors by vehicle type, age, fuel, engine, and pollutants to individual daily total VKT to compute the hot exhaust emissions. The rest of the processes for cold start and evaporative emissions are calculated afterwards. The emission calculation methodologies used in the CARS model are based on tier 2 and tier 3 methodologies from the EEA's mobile emission inventory guidebook (EEA, 2019) to be consistent with Korea's National Emission Inventory System (NEIS) (Lee et al., 2011a).

### 2.2.1 Hot Exhaust Emissions

Hot exhaust emission, which is from the vehicle's tailpipe, is the exhaust gas from the combustion process in an ICE. The ICE combustion cycle generally causes incomplete combustion processes which emit hydrocarbons, carbon monoxide (CO), and particulate matter (PM) into the atmosphere. The sulfur compounds in the fuel are oxidized and become sulfur oxides ($SO_x$). Nitrogen oxides ($NO_x$) are similarly produced during the combustion process due to the abundant nitrogen ($N_2$) and oxygen ($O_2$) in the atmosphere.

Equation 3 represents the calculation of daily individual vehicle hot exhaust emission rate, $E_{hot;p,v,age}$ (g d$^{-1}$) of pollutant ($p$). An individual vehicle-specific daily $VKT_{v,age}$ (km d$^{-1}$) is estimated by Eq. (1). The $EF_{hot;p,v,age,s}$ (g/km) is the hot exhaust emission factor of pollutants ($p$) for the vehicle type ($v$), vehicle age ($age$), and average vehicle speed ($s$). The district's total emission rate is the total hot exhaust emissions from all individual vehicles within the same district.

$$E_{hot;\,p,v,age} = DF_{p,v,age} \times VKT_{v,age} \times EF_{hot;\,p,v,age,s} \qquad (3)$$

The deterioration factor ($DF$) in Eq. (3) is an optional function in the CARS model that can be turned on or off by users. This deterioration process is caused by vehicle aging and can lead to the increase of vehicle emissions. The CARS model applies the vehicle registration year to estimate the deterioration factor as additional emissions, which vary by vehicle type and pollutant. According to the guidance of deterioration factors calculation from NIER, there is no deterioration in a new vehicle in their first five years. After five years, the deterioration factors can increase the range by 10% depending on the type of vehicle and pollutants. Deterioration processes can cause a 50% or 100% increase of emissions in fifteen-year-old vehicles. Currently, the $DF$ is an empirical coefficient that varies by vehicle age (Lee et al., 2011a).



The hot exhaust emission factor, $EF_{hot;p,v,s}$ (g/km) is a function of vehicle speed ($s$) with
other empirical coefficients: $a, b, c, d, f, k$. The emission factor formula and those coefficients were
developed by NIER CAPSS (Lee et al., 2011a). These coefficients are varied by pollutants ($p$),
vehicle type ($v$), vehicle age ($age$), and vehicle speed ($s$). The vehicle speed affects the combustion
efficiency of an ICE and impacts the emission rates and its composition from the tailpipe.
$$EF_{hot;p,v,age,s} = k(a \times s^b + c \times s^d + f) \qquad (4)$$
While vehicle speed plays a critical role in hot exhaust emissions from most vehicles, NOx
emissions from some diesel vehicles show sensitivity to local ambient temperature along with
vehicle speed (Ntziachristos and Samaras, 2000). Figure 3 shows the dependency of $NO_x$ emission
factors from compact diesel vehicles to vehicle speed (Fig. 3a) and ambient temperature (Fig. 3b).
Figure 3a shows a significant decrease of $NO_x$ emissions while speed increases. Figure 3b
demonstrates the significance of local meteorology on $NO_x$ emissions from a compact diesel sedan.
Based on these NIER's CAPSS emission factors, the sensitivity to local ambient temperature is
limited to $NO_x$ pollutant emissions from diesel vehicles.
Due to its high sensitivity to the vehicle operating speed, it is important for the CARS to
simulate realistic speed patterns for accurate emissions estimates. When a constant single speed is
assigned to compute hot exhaust emissions, it won't reflect the emissions under low-speed
circumstances, which could cause higher emissions due to its incomplete ICE combustion. To
overcome this limitation, the CARS has adopted the 16 average speed bins concepts for a better
representation of vehicle speed distribution that varies by road type (i.e., local, highway,
expressway). We have implemented a feature for the CARS optionally to apply road-specific
average speed distributions (ASD) ($A_{bin,r}$), which represents the fractions of 16-speed bins ($bin$)
(from 0 to 121 km h$^{-1}$ defined in Appendix E) for eight different road types ($r$) (No.101-108, shown
in Appendix C) as classified by CAPSS (Fig. 4). Although ASD patterns vary by region, we did
not implement the regional variations of ASD due to the lack of region-specific vehicle speed
measurements in South Korea.
In this study, we developed the most realistic ASDs for eight different road types (No. 101-
108) in South Korea based on the latest road link-specific average speed and AADT from the GIS
road network shapefiles (NIER, 2018) and the U.S. EPA's MOVES ASD datasets (USEPA, 2020).
Because a single average speed was assigned to each road link, the ASDs based on South Korea's
GSI road shapefiles did not capture the low-speed range (<16 km h$^{-1}$) that occurs in reality.
Therefore, we incorporated the ASD developed by U.S. EPA with Georgia state ASD to improve
the representation of the low-speed range (speed bin #1 and #2). We modified the total fractions
of low-speed bins (the 2:1 ratio of fractions of bin #1 and #2) by adding 2% of distribution for
interstate expressways, 3% of distribution for urban expressways, 7% of distribution for all
highways, and 15% for all local roads. Further, those increases of low-speed bins reduced the

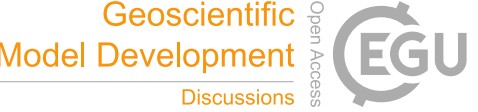

distributions of other higher speed bins homogeneously due to the renormalization of fractions by
road type. Figure 4 shows the renormalized ASDs of all road types applied in this study.
While 16-speed bins ASD application is critical to computing more realistic hot exhaust
emissions, there should be some restrictions on certain road types. Users can adjust the restricted
roads control table input file to limit the vehicle types that can only be operated on a particular
road type. For example, motorcycles are limited to local roads (No. 104, 106, and 107), but not on
expressways (No. 101, 102, 103, 105, and 108) due to its traffic regulation rules. Heavy trucks are
only allowed on the highway (No. 101, 102, 103, 105, and 108.) by law. The details of the road
restriction control table format can be found on the CARS's user's guide from the CARS Github
website (https://github.com/bokhaeng/CARS/tree/master/docs/User_Manual).
The 16-speed bins averaged speed distribution calculated by road type ($A_{bin,r}$) and road type
weight factors ($\omega_{r,d}$) in a district ($d$) from Eq. (13) are added to the CARS hot exhaust emissions
equation (Eq. 3). The hot exhaust emissions from individual vehicles ($E_{hot;p,v,age}$) can be calculated
by considering road-specific speed bins distribution (Eq. 5). Although the vehicles may be
operated in different districts from their registered district, this is our best method to estimate the
vehicle speed for hot exhaust emissions.
$$E_{hot;\,p,v,age} = DF_{p,v,age} \times \sum_{bin}(VKT_{v,age} \times EF_{hot;\,p,v,age,s} \times A_{bin,r}) \qquad (5)$$

**2.2.2 Cold Start Emissions**
The cold start emissions occur when a cold-engine vehicle is ignited. The lower
temperature of the ICE is not an optimal condition for complete fuel combustion. This process
lowers the combustion efficiency (CE) and increases the emissions of hydrocarbon and CO
pollutants from the tailpipe exhaust (Jang et al., 2007). The CARS can estimate the cold start
emissions for vehicles using gasoline, diesel, or liquefied petroleum gas (LPG) fuel. Besides the
vehicle and engine type, road type also plays a critical role in the quantity of cold start emissions
because it occurs mostly in parking lots and rarely on highways.
The cold start emission, $E_{cold}$ (g d$^{-1}$), is derived from the hot exhaust emissions, the ratio of
hot to cold exhaust emissions ($EF_{cold}/EF_{hot}$ -1.0), and the percentage of the traveled distance with
a cold engine (Eq. 6).
$$E_{cold;\,p,v} = \beta_T \times E_{hot;\,p,v} \times \left(\frac{EF_{cold;\,p,v}}{EF_{hot;\,p,v}} - 1.0\right) \qquad (6)$$

The emission factor of cold start emissions ($EF_{cold}$) is not directly calculated from
measurement data like hot exhaust emissions ($E_{hot;p,v}$), but measured under different ambient
temperatures ($T$). The CARS model applies linear regression models developed by CAPSS to
estimate the increasing ratio of cold start to hot exhaust emissions ($EF_{cold}/EF_{hot}$) under different



temperatures ($T$) (Eq. 7). In this equation, $A$ and $B$ are the empirical coefficients that vary by the
pollutants ($p$) and vehicle type ($v$).
$$\left(\frac{EF_{cold;\,p,v}}{EF_{hot;\,p,v}}\right) = A_{p,v} + B_{p,v} \times T \qquad (7)$$
$\beta$ is the percentage of the distance traveled under a cold engine. It also depends on the
ambient temperature. Cold ambient temperatures cause a longer distance traveled under a cold
engine due to the slower heating time. According to the CAPSS database for Seoul city (Lee et al.,
2011a), the empirical linear equation for $\beta$ is shown in Eq. (8). This formula represents how
ambient temperature affects $\beta$. For example, when the average temperature is -2°C, $\beta$ is 34.8%.
In summer, the monthly average temperature is 25.7°C, which causes $\beta$ to drop to 21%.
$$\beta = 0.647 - 0.025 \times 12.35 - (0.00974 - 0.000385 \times 12.35) \times T \qquad (8)$$
**2.2.3   Evaporative VOC Emissions**
Evaporative emissions are emissions from vehicle fuel that are evaporated into the
atmosphere. This occurs in the fueling system inside the vehicle, such as fuel-tanks, injection
systems, and fuel lines. Diesel vehicles, however, can be exempted due to diesel fuel's low vapor
pressure. The primary sources of evaporative emissions are breathing losses through tank vents
and fuel permeation/leakage. The CARS model adopted the EEA's emission inventory guidebook
(EEA, 2019) to account for three mechanisms to estimate the evaporative VOC emissions ($E_{vap}$):
diurnal emissions from the tank ($e_d$), hot and warm soak emissions by fuel injection type ($S_{fi}$), and
running loss emissions ($R$) (Eq. 9). Unlike CAPSS, there is a conversion factor (0.075) applied to
$E_{vap}$ for motorcycles to prevent an over-estimation of VOC.
$$E_{vap;\,p,v} = \left(e_{d;\,p,v} + S_{fi;\,p,v} + R_{l;\,p,v}\right) \qquad (9)$$
Diurnal emissions, $e_d$ (g d$^{-1}$), during the daytime are caused by the ambient temperature
increase and the expansion of fuel vapors inside the fuel tank. Most of the current fuel tank systems
have emission control systems to limit this kind of evaporative VOC emissions. The $e_d$ can be
calculated with the empirical Eq. (10), which was developed by CAPSS. $T_l$ is the monthly average
of the daily lowest temperatures and $T_h$ is the monthly average of the daily highest temperatures.
The empirical coefficient α is 0.2, which represents how 80% of emissions are eliminated by the
vehicle emission control system.
$$e_d = \alpha \times 9.1 \, exp[0.3286 + 0.0574 \times (T_l) + 0.0614 \times (T_h - T_l - 11.7)] \qquad (10)$$
Soak emissions ($S_{fi}$) occur when a hot ICE is turned off; the remaining heat from the ICE
can increase the fuel temperature in the system. The carburetor float bowls are the major source of



45  the soak emissions. Newer vehicles with fuel injection and return-less fuel systems do not emit
soak emissions. Because most of the current vehicles in South Korea have a new fuel system, soak
emissions ($S_{fi}$) in the CARS model are set to 0.
The running loss emissions ($R_l$) are from vapors generated in the fuel tank when a vehicle
is in operation (Eq. 11). In some older vehicles, the carburetor and engine operation can increase
the temperature in the fuel tank and carburetor, which can cause a significant increase in
evaporative VOC emissions. VOC emissions from running loss can be greatly increased during
warmer weather. However, newer vehicles with fuel injection and return-less fuel systems are not
affected by the ambient temperature. Because most vehicles in South Korea do not use carburetor
technology, we expect running loss emissions to have the least impact (Lee et al., 2011b).
$$R_l = \alpha \times L_{r,v} \times [(1 - \beta) \times R_h + \beta \times R_w] \qquad (11)$$
The empirical coefficient $\alpha$ is 0.1 here, which represents that 90% of the running loss is
avoided by the newer fuel system. $L$ is the distance traveled (km) by road and is the same one used
in hot exhaust emission calculations. $\beta$ is the same parameter from Eq. (8). The $R_h$ and $R_w$ are the
average emission factors from running loss under hot and warm/cold conditions, respectively.
**2.3    Road Link-Level Emissions Calculations**
In general, district-level automobile emissions calculations are driven by district-level
averaged vehicle activity and operating data, which do not reflect realistic spatial patterns of
onroad automobile emissions.  The CARS model introduces road link-specific traffic data by
default to develop spatially enhanced road link-specific emissions that reflect more representative
emissions by road link. This high-resolution traffic data is a GIS shapefile that is composed of
many connected segments, which are called "road links." All road links hold information such as
start/end location coordinates, AADT, road link length, averaged vehicle speed, and road type (No.
101-108).

The CARS model applies link-level AADT ($AADT_{d,r,l}$, d$^{-1}$) and road length ($L_{d,r,l}$) to
compute the road link-specific VKT ($VKT_{d,r,l}$, km d$^{-1}$) in Eq. (12). The road links are identified by
district ($d$), road type ($r$), and link ($l$) labels. The road VKT is a parameter that reflects the traffic
activity of each road link and it is different from individual daily vehicle activity data ($VKT_{v,age}$)
in Eq. (1).
$$VKT_{d,r,l} = AADT_{d,r,l} \times L_{d,r,l} \qquad (12)$$
Road link-specific VKT ($VKT_{d,r,l}$) is used to redistribute the district total emissions ($E_{onroad}$)
from Eq. 2 into road link-level emissions. The following three weight factors are computed: the
district weight factors, $\omega_d$ (Eq. 13), the road type weight factors, $\omega_{d,r}$ (Eq. 14), and the road-link





378 weight factors, $\omega_{d,l}$ (Eq. 15). The weight district factors ($\omega_d$) are the renormalization of each
379 district's total VKT over state-level total VKT ($N$ is the number of districts). The main reason we
380 performed the renormalization over state-level total VKT is to reflect daily traffic patterns from
381 multiple districts under the assumption that most vehicles travel within the same state. The road
382 type weight factors by district ($\omega_{r,d}$) are used to compute road-specific emissions, while road-
383 specific averaged speed distributions (ASD; $A_{s,r}$) from Eq. (5) are applied to capture vehicle
384 operating speeds by road type. The road link weight factors ($\omega_{d,l}$) are then applied to redistribute
385 the district emissions into road link-level emissions.


$$\omega_d = \frac{\sum_r \sum_l VKT_{d,r,l}}{\frac{1}{N}\sum_d \sum_r \sum_l VKT_{d,r,l}} \qquad (13)$$

$$\omega_{d,r} = \frac{\sum_l VKT_{d,r,l}}{\sum_r \sum_l VKT_{d,r,l}} \qquad (14)$$

$$\omega_{d,l} = \frac{VKT_{d,r,l}}{\sum_r \sum_l VKT_{d,r,l}} \qquad (15)$$

390 **3 CARS Configuration**

391  The CARS model is an open-source program based on Python (Guido van Rossum, 2009)
392 that allows the users to efficiently apply open-source modules to develop programs. Users can
393 easily install Python development tools and load customized packages and modules to set up the
394 CARS development environment. All CARS modules are developed using Python v3.6. Other than
395 the GIS road shapefiles, all input files are based in the ASCII CSV format, which can be easily
396 handled by both spreadsheet programs and programming languages, making it more accessible for
397 users of all skillsets. The CARS can not only estimate district-level and spatially enhanced road
398 link-level emissions, but can also generate hourly chemically speciated gridded emissions for
399 CTMs. In addition, the CARS also generates various summary reports, graphics, and
400 georeferenced plots for quality assurance.

401  The required Python modules for the CARS are: "***geopandas,***" "***shapely.geometry***", and
402 "***csv***" modules to read the shapefiles and table data files. The "***NumPy***" and "***pandas***" modules
403 are used to operate the memory arrays and scientific calculations while the "***pyproj***" module deals
404 with converting the projection coordinate systems. "***matplotlib***" is for generating any type of
405 figures/plots. Furthermore, the CARS model can also read and write Climate and Forecast (CF)-
406 compliant NetCDF-formatted files using "***NetCDF4***".

407  The first process in the CARS is "***Loading_function_path***"; it allows users to define and
408 check the input file paths. Once all input files are checked, there are six process modules in CARS
409 to process inputs, compute emissions, and generate various output files, including QA reports.



Figure 5 is the schematic of the CARS that consists of six process modules with various functions. The six process modules are (1) "**Process activity data**", (2) "**Process emission factors**", (3) "**Process shapefile**, (4) "**Calculate district emissions**", (5) "**Grid4AQM"**, and (6) "**Plot figures**". The main purpose of modularizing the CARS is to meet the needs of various communities, such as policymakers, stakeholders, and air quality modelers. While modules (1) through (4) are required to develop the district-level and road link-level emissions inventories, module (5) "**Grid4AQM**" is optional depending on if users want to develop chemically-speciated gridded hourly emissions for CTMs. Also, the modularity system in the CARS allows users to bypass certain modules if it has been previously processed without any changes. For example, if there is no change in traffic activity, emission factors table, or GIS shapefiles, users do not need to run these modules and can simply read the data frame outputs and then run "**Grid4AQM**" for the modeling dates and domain. The "**Grid4AQM**" module will not only improve the computational time for CTMs but also eliminate the need for a $3^{rd}$ party emissions modeling system like SMOKE (Baek and Seppanen, 2021).

The rectangle boxes in Fig. 5 represent the data array and the boxes with rounded edges are the functions in the CARS. Details on the CARS code, input table format, and functions setup information can be found on the CARS GitHub website (Pedruzzi *et al.*, 2020).

The "**Process activity data**" module first reads the vehicle activity data, such as an individual vehicle's daily total VKT based on its registered district. The "**Process emission factors**" module reads and stores the emission factors table that holds all pollutant emission factors to estimate the emissions for all vehicles. Meteorology-sensitive emission factors are only limited to $NO_x$ pollutants. District boundary GIS shapefiles and road network shapefiles are processed through "**Process shape file**" to generate the VKT-based redistribution weighting factors from Eq. (13), (14) and (15) for the "**Calculate district emissions**" module to compute district-level and road link-level emission rates (metric tons per year, t $yr^{-1}$).

The redistributed emission rates (t $yr^{-1}$) from the "**Calculate district emissions**" module present annual total emission rates until district-level VKTs from the "**Process activity data**" module are added. Then, the "**Grid4AQM**" module can generate CTM-ready chemically speciated emissions. The "***Read_chemical***" function from the "**Grid4AQM**" module is designed to process the chemical speciation profile that can convert the inventory pollutants such as CO, $NO_X$, $SO_2$, $PM_{10}$, $PM_{2.5}$, VOC, and $NH_3$, into the chemically lumped model species that CTM requires for chemical mechanisms, such as SAPRC (L. and Heo, 2012) and Carbon Bond version 6 (CB6) (Yarwood and Jung, 2010). The "***Read_temporal***" function processes the complete set of monthly, weekly, and hourly temporal allocation profiles that can convert annual total emissions to hourly emissions. "***Read_griddesc***" defines the CTM-ready modeling domain and computes the gridding fractions for all road link-level emissions by overlaying the modeling domain over the GIS shapefiles. Once annual total emissions are chemically speciated, spatially gridded, and temporally allocated into hourly emissions, the "***Gridded_emis***" function will combine emission source-level



conversion fractions from each function (***Read_chemical***, ***Read_temporal***, and ***Read_griddesc***) to
generate the CTM-ready chemically speciated, gridded hourly emissions in the NetCDF binary
format. The "**Plot Figures**" module is designed for generating various summary reports and
graphics to assist users in understanding the estimated automobile emissions inventory computed
by the CARS. The following section will describe the detailed processes of the "**Grid4AQM**"
module, which includes chemical, spatial, and temporal allocations.

### 3.1    Chemical Speciation

To support CTMs applications, the CARS needs to be able to convert inventory pollutants
into chemical lumped model species based on the choice of CTM chemical mechanisms. $NO_x$
includes nitric oxide (NO), nitrogen dioxide ($NO_2$), and nitrous acid (HONO). VOCs can represent
hundreds of different organic carbon species, such as benzene, acetaldehyde, and formaldehyde.
These grouped inventory pollutants cannot be directly imported into the chemical mechanism
modules in the CTM system and require chemical speciation allocation for CTMs to process them
during their chemical reactions. Therefore, the "**Grid4AQM**" module performs the chemical
species allocation step prior to the temporal and spatial allocations to generate the gridded hourly
emissions. The "***Read_chemical***" function in "**Grid4AQM**" module allows users to assign these
emission inventory pollutants to CTM-ready surrogate chemical species (a.k.a lumped chemical
species) by vehicle, engine, and fuel type. For example, VOC emissions from diesel busses can be
converted into the following composition based on its chemical allocation profile: alkanes (68%),
toluene (9%), xylenes (8%), alkenes (4%), ethylene (2%), benzene (1.3%), and unreactive
compounds (7%) when CB6 chemical mechanism is selected. Further details on the chemical
speciation profile input formats are available in the CARS user's guide.

### 3.2    Spatial Allocation

The "**Calculate district emissions**" module calculates not only the total district emissions
but also road link-specific emissions based on road link-specific AADT data from road network
GIS shapefiles. The "**Calculate district emissions**" module first gets the district total vehicle
emissions (Eq. 2) based on the district-level VKTs, and then the normalized district total emissions
by district weight factor, $\omega_d$ (Eq. 13). Afterwards, the normalized district total emissions are
redistributed into every road link using road link-level weight factors ($\omega_{d,l}$) (Eq. 15). The district
total emissions from Eq. (2) and from Eq. (15) remain the same. Then the computed road link-
level emissions then will be converted into grid cell emissions using the modeling domain grid cell
fractions computed in the "***Read_griddesc***" function in the "**Grid4AQM**" module.





### 3.3 Temporal Allocation


Once chemical and spatial allocations are completed, the final step to support CTM
application is a temporal allocation that converts the annual total emissions from the "**Calculate**
**district emissions**" module into hourly emissions. The "***Read_temporal***" temporal allocation
function in the "**Grid4AQM**" module converts the annual emission rate (t yr$^{-1}$) to the hourly
emission rate (mol hr$^{-1}$) using monthly, weekly, and weekday/weekend diurnal temporal profiles.
This module processes these temporal profile inputs, which are the monthly (January - December),
weekly (Monday - Sunday), and weekday/weekend 24 hour profile tables (0:00-23:00 LST). The
users can assign these temporal profiles with a combination of vehicle, engine, fuel, and road types
to enhance their temporal representations in detail.

### 3.4 Chemical Transport Model Emissions


The main goal of the **"Grid4AQM"** module is to generate temporally, chemically, and
spatially enhanced CTM-ready gridded hourly emissions. First, it reads the CTM modeling domain
configuration and then overlays it over the road network GIS shapefile and district-boundary
shapefile to define the modeling domain. This overlaying process between the road network,
district boundary GIS shapefiles, and modeling domain allows the **"Grid4AQM"** module to
compute the fraction of road links that intersects with each grid cell. Figure 6 demonstrates how
the district boundary and road network GIS shapefiles are used to perform the spatial allocation
processes in CARS. Figure 6a is a native road link shapefile of Seoul with AADT, VKT, district
ID, and road type. Figure 6b presents an overlay of two districts's road links (purple and blue)
over the selected region. State total emissions will be renormalized into weighed district total
emission data and then redistributed into the road link. Figure 6c illustrates how the weighted road
link-level emissions get allocated into modeling grid cells for CTMs. The link-level VKT ($VKT_{d,r,l}$)
from Eq. (12) will be used to compute a total of traffic activity fractions by grid cell and then use
that to assign the link-level emissions from Eq. (2) into each grid cell. When a road link intersects
with multiple grid cells, the **"Grid4AQM"** module will weigh the emissions by the length of the
link that intersects with each grid cell.

Through the overlay process, the CARS model can generate various types of output data,
such as total district emissions, link-level emissions, and CTM-ready gridded emissions. For
example, the CO vehicle emissions from the Seoul metropolitan in South Korea are presented in
three different output formats in Fig. 7. Figure 7a shows the annual mobile PM$_{2.5}$ emissions by
district. The road link level annual emissions are presented in Fig. 7b. Furthermore, the CARS
applies the link-level emissions from Fig. 7b to generate the hourly grid cell emission data with a
1 km × 1 km resolution for the CTM in Fig. 7c.

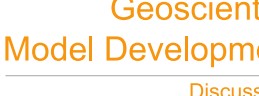
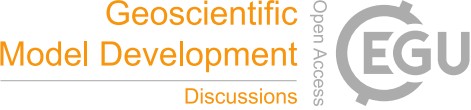

### 3.5 National Control Strategy Application


One of the unique features in the CARS compared to other mobile emissions models is that
it can promptly develop controlled mobile emissions responding to the national emergency high
PM$_{2.5}$ episodes. It is very common to experience high PM$_{2.5}$ episodes, especially during the
wintertime in South Korea due to domestic and international primary and secondary air pollutants
emissions. When the 72 hour forecasted PM$_{2.5}$ concentration exceeds the average 50 µg/m$^3$ (0:00-
16:00 LST), the national PM$_{2.5}$ emergency control strategy is activated for ten days. It applies a
nationwide vehicle restriction policy within 24 hours. It enforces a limit on what kind of vehicles
can be operated on a certain date. The restrictions can be applied in the following ways: the
closures of public parks and government facilities, and restrictions of certain vehicles based on
their fuel type and age, which is a major factor of engine deterioration. This policy will limit the
number of vehicles on the network roads significantly, which could reduce primary PM$_{2.5}$ and
precursor pollutant (NOx, NH$_3$, and VOC) emissions, especially from heavily populated
metropolitan regions (Choi et al., 2014; Kim et al., 2017a; Kim et al., 2017b; Kim et al., 2017c).
To understand the impacts of an even/odd vehicle restriction policy in real-time, we need to
quickly develop a rapid control response emissions for the air quality forecast modeling system.
The process of generating the controlled mobile emissions can take a long time if we start fresh.
Thus, we have implemented this control strategy as an optional "***Control Factors***" function in the
"**Calculate district emissions**" in the module for users to quickly and easily generate the
controlled mobile emissions with consideration of the limited number of vehicles based on the
vehicle, engine, fuel, and vehicle manufactured year. A one hundred percent (100%) control factor
means that there are no emissions from those selected vehicles.
Because of the modularization system in the CARS, we can bypass some computationally
expensive data processing modules (i.e., "**Process activity data**", "**Process emission factors**",
and "**Process shape file**") and let the "**Calculate district emissions**" module quickly apply control
factors while it computes the district-level mobile emission inventory from Eq. (2). This will allow
users to reduce the computational time to generate the controlled mobile emissions under a specific
control scenario and develop the controlled CTM-ready gridded hourly emissions using the
"**Grid4AQM**" module.

### 3.6 Computational Time


While the CARS can generate a high-quality spatiotemporal emission inventory for
policymakers, stakeholders, and air quality modelers, it is quite critical for the CARS to generate
these complex mobile emissions effectively and accurately without being at the expense of
computational time. This is especially important to meet the needs for an air quality forecast
modeling system responding to a national emergency control strategy implementation.



In this section, we will discuss the details of the CARS computational modeling performance. While the CARS model has been highly optimized, the modularization of CARS has also improved its modeling performance with optional module runs. The breakdown of module-specific computational time estimates based on the benchmark CARS runs are listed in Table 1. The benchmark CARS case includes a total of 24,383,578 daily VKT datasets from KSTA over two different years, 84,608 emission factors for all pollutants across a combination of vehicle-age-engine-fuel types, 385,795 road links from the GIS road network shapefiles, 5,150 districts/16-states boundary GIS shapefile, and 5,494 grid cells (=82 rows and 67 columns) for CTMs. Without any computational parallelization, the total processing time of all six modules usually takes around a half hour to generate a single day CTM-ready gridded hourly emission file. However, it can be further shortened to 25-30 minutes on a higher performance computer. Because of the modular system implemented in the CARS, generating one month (31 days) long gridded hourly emissions from CTMs do not require over 15 computational hours, but only around 100 minutes on high-performance computers. The maximum usage of RAM can reach up to 11 GB. Table 1 shows the breakdown of computational time by each module from two different hardwares (desktop and laptop computers). The numbers in parentheses beside the **"Grid4AQM"** module is the computational time for a single day versus 31 days. While the **"Grid4AQM"** module takes an average of 4.9 minutes for a single day emissions generation, processing a consecutive 31 days saves 46% more time, decreasing from 151.9 minutes (=4.9 minutes * 31 days) to 81.6 minutes.

## 4 Results

### CARS and CAPSS Comparison

The CARS model calculates the 2015 onroad automobile emissions based on the latest 2015 emission factors and the 2015-2017 vehicle activity database in South Korea. The annual total emissions from CARS are compared against the ones from NIER CAPSS in Table 2. The CARS model estimated the following annual total emissions in units of metric tons per year (t $yr^{-1}$): $NO_x$ (301,794); VOC (61,186); CO (373,864), $NH_3$ (12,453); $PM_{2.5}$ (10,108), and $SO_x$ (172.0). Compared to NIER CAPSS, the CARS overestimated all pollutants except for $NO_x$ (-18% decrease) and $SO_x$ (-17% decrease). It overestimated the emissions of VOC by 33%, $PM_{2.5}$ by 15%, CO by 52%, and $NH_3$ by 24%. Both NIER CAPSS and CARS shared the same emission factor tables, which hold over 84,608 emission factors for all pollutants across a combination of vehicle, age, engine, and fuel types.

The difference between CAPSS and CARS approaches are caused by three reasons: First, the number of vehicles used in CARS is slightly higher (6%) than CAPSS data (1.3 out of 23 million), as well as other key traffic-related activity inputs (i.e., vehicle age distribution, averaged speed distribution, etc). Secondly, the vehicle speed information assigned by vehicle and road type play a critical role in the differences between CAPSS and CARS. The CAPSS calculation was



based on the road-specific mean speed value or 80% of the speed limit as an input of vehicle
operating speed by three road types (rural, urban, and expressway). In other words, CAPSS only
assigns a "single-speed value" for each road type, and does not encounter the variation of vehicle
speed during its operation on roads into the emissions calculation. Most running exhaust emissions
occur during a vehicle's low-speed operation due to its incomplete combustion of fuel, and it is
critical to accurately represent the emissions across various speed bins in order to compute the
correct emissions. The CARS model has an option to apply the average speed distribution (ASD)
over 16 speed bins for eight road types (Fig. 4). The CARS speed distribution process can better
represent the speed variations of vehicle speeds for each road type. A detailed analysis of the
impact of vehicle speed will be discussed later in this chapter. Lastly, other advanced processes in
the CARS, such as link-level AADT and district-level vehicle data (5,150 districts in South Korea),
can reflect more spatial detail and variation than the CAPSS. The CAPSS only considers state-
level data (17 states in South Korea) and five road types (interstate expressway, urban highway,
rural highway, urban local, and rural local).
Figure 8 illustrates more details about the difference between the annual emissions from
CARS to the CAPSS by pollutants and vehicle types. Sedan vehicles show the largest increase of
VOC (33%), CO (41%), and $NH_3$ (23%) in the CARS relative to CAPSS because almost 56% of
total vehicle count (13.5 million) is composed of sedan vehicles. Also, sedan vehicles contribute
51% of total VOC and 61% of total CO annual emissions. The VOC and CO emissions from sedans
are largely affected by the average speed distribution process when compared to other vehicle
types. Similarly, the largest decreases of $NO_x$ (-16%) and $SO_x$ (-18%) are from trucks because they
are significant $NO_x$ (~50%) and $SO_x$ contributors (~27%) and their emission factors are sensitive
to vehicle speed.
**Onroad Emissions Analysis**
The CARS is a bottom-up emissions model, which utilizes local individual vehicle activity
data, detailed local emission factors for every vehicle and fuel type, and localized inputs such as
average speed distribution by road type and deterioration factor. It allows users to assess the
detailed breakdown of localized emission contributions. Table 3 represents the individual air
pollutants ($NO_x$, VOC, $PM_{2.5}$, CO, $NH_3$, and $SO_x$) emission contributions (t $yr^{-1}$), fractions (%),
and impact factors (IF) by the vehicle type and fuel system. The IF is defined by the normalized
annual emissions with vehicle counts of each category (kg $yr^{-1}$ per vehicle). The CARS also can
provide the average daily VKT per vehicle, which is the total daily VKT divided by vehicle
numbers, to explain the emission contributions in Appendix D.
Diesel-fueled vehicles contribute the most of $NO_x$ emissions, which is over 85.3% (257,305
t $yr^{-1}$), although the number of diesel vehicles only amounts to approximately 35% of the total
vehicles (Table 3a). While the diesel trucks emitted 49.1% (148,246 t $yr^{-1}$) of total $NO_x$ with an IF



value of 47.9 (kg yr$^{-1}$), the highest impact (IF = 340 kg yr$^{-1}$) occurred from diesel buses with only a 8.51% contribution to the total NOx emissions. This is caused by the highest average daily VKT from diesel buses compared to other vehicles, which is expected in a highly populated metropolitan area like Seoul, South Korea. A diesel bus generally has a 3-5 times higher daily VKT (180 km d$^{-1}$) than other common vehicles (gasoline sedan: 34 km d$^{-1}$, diesel truck: 57 km d$^{-1}$). The second-largest vehicle type is the CNG (compressed natural gas) bus (248 kg yr$^{-1}$), which also has a higher VKT. Their average daily VKT is 212 km d$^{-1}$, with only a 3.1% NO$_x$ contribution.

For VOC emissions, over 12 million gasoline vehicles cause 52.1% (31,885 t yr$^{-1}$) of the total VOC emissions, and the gasoline sedan is the highest contributor across all vehicle types, which is over 28,434 t yr$^{-1}$ (46.5%) (Table 3b). Unlike NOx emissions, diesel vehicles only contribute 23.0% (14,070 t yr$^{-1}$) of the total VOC emissions. Across the vehicle fuel types, the IF outcome indicates that CNG vehicles have the highest IF values for VOC, which is 247 kg yr$^{-1}$ due to the relatively high VOC contribution (19% over total VOC) and a low number of heavy CNG vehicles. The IF of CNG trucks are 77.2 kg yr$^{-1}$, but only contribute 0.2% to total VOC emissions. The IF of the CNG bus is 320 kg yr$^{-1}$ and emits 19.5% of the total VOC. Comparing the IFs of buses across fuel types, the CNG bus emits less NO$_x$ but higher VOC than a diesel vehicle. Each CNG bus has about 33 times higher IF of VOC (320 kg yr$^{-1}$) than a diesel bus (9.51 kg yr$^{-1}$), and CNG buses released slightly lower NO$_x$ (248 kg yr$^{-1}$) than diesel buses (340 kg yr$^{-1}$) (Table 3a and 3b). It indicates that a CNG bus is better for rural areas and a diesel bus is better for urban areas to control ozone, because the rural area is usually NO$_x$ limited for ozone formation and urban areas are VOC limited.

The current South Korea CAPSS onroad emissions inventory does not consider the PM$_{2.5}$ emissions from tire and brake wear, which are the highest contributors of PM$_{2.5}$ emissions from vehicles on roads. For that reason, diesel vehicles become the major source of PM$_{2.5}$ emissions, which contributes over 98.5% (9,959 t yr$^{-1}$) of the PM$_{2.5}$ emissions based on the CARS 2015 emissions (Table 3c). The diesel truck, SUV, and van are the three major sources, and their contributions of total PM$_{2.5}$ are 53.6%, 21.4%, and 11.2%, respectively. Although over 52% of the vehicles are gasoline vehicles, their primary PM$_{2.5}$ contribution is limited to 1.44%. The diesel bus has the highest IF (2.83 kg yr$^{-1}$), which is caused by the largest average daily VKTs.

Similar to VOC emissions, CO is mostly emitted through the tailpipe due to incomplete internal combustion of fuel and share similar emissions distributions across vehicle and fuel types (Table 3d). Gasoline vehicles contribute most of the CO (220,390 t yr$^{-1}$, 59.0%), and sedan vehicles are the primary source (178,121 t yr$^{-1}$, 47.6%) of this out of all gasoline vehicles. Across vehicle types, bus shows the highest IF of CO (81.2 kg yr$^{-1}$) due to its largest daily VKT. CO is the most abundant pollutant released from vehicles (373,864 t yr$^{-1}$) across all pollutants from onroad automobile sources. Although CO is much less reactive than other vehicle VOCs (Rinke and Zetzsch, 1984; Liu and Sander, 2015), the majority of CO emissions from onroad automobile sources plays a critical role in generating 30% of hydroperoxyl radicals (HO$_2$) and causing ozone



formation in urban areas (Pfister et al., 2019). Thus, CO is also another crucial precursor to ozone
formation in urban areas.
$SO_x$ emissions are related to the sulfur content within the fuel component; diesel has a
higher sulfur content than any other fuels. Most $SO_x$ is contributed by diesel vehicles (93.8 t $yr^{-1}$,
54.5%) (Table 3e). Within diesel vehicles, trucks provide 26.5% of $SO_x$ (45. t $yr^{-1}$). Although the
$SO_x$ from sedan vehicles are slightly higher (~3.3%) than diesel trucks, the number of diesel trucks
is only 29.6% of the number of gasoline sedans. Thus, diesel trucks have a higher IF than gasoline
sedans. Across vehicle types, buses have the highest IF (0.095 kg $yr^{-1}$) of $SO_x$, and diesel buses in
particular have the largest IF at 0.143 kg $yr^{-1}$.
The $NH_3$ emissions table (table 3f) indicates that 98.7% of $NH_3$ is from gasoline vehicles
while diesel trucks only contribute 1.13%. The IF result also shows that the gasoline sedan has the
most significant impact per vehicle (1.17 kg $yr^{-1}$).
According to the vehicle activity and the CARS model results, nearly half of the total
vehicles (24.3 million) are gasoline sedans (10.4 million, 42.8%), and gasoline sedan vehicles
contributed most of the VOC and CO emissions (46.5% and 47.6%), but only 7.7% of the total
$NO_x$ emissions. The number of diesel vehicles is 8.6 million (35.4%); however, they emitted about
85.3% of the total $NO_x$ and 98.5% of the primary $PM_{2.5}$. These results indicated that the annual
traffic-related mobile emissions are not only affected by the number of vehicles, but also by
different vehicle and fuel types. Therefore, this study normalized the annual emissions by the
number of vehicles to confirm the emission composition by individual vehicle types.
**Average Speed Impact Study**
The CARS can also optionally apply the average speed distribution (ASD) by road type to
compute more realistic mobile emissions on the road network when compared to using a current
single average speed value for each road type (Appendix E). Applying the ASD will generate a
much better representation of actual traffic patterns from each road type. To understand the impacts
of ASD application, we performed sensitivity runs between using a single-speed to the ASD
application (Appendix F). The ASD data was described in Fig. 4, and the road-specific average
single-speed values were developed based on the weighted average method using the same ASD
data. Appendix E and S6 describes the details of ASD as well as road-specific speed values.
Figure 9a shows the differences in total emissions between two scenarios and is organized
by pollutant. The single-speed scenario largely underestimates the emissions across all pollutants
compared to the ones from the ASD scenario. $NO_x$ (16%), VOC (40%), and CO (30%) were
especially underestimated. The difference is caused by the lack of low-speed bins (<16 km $h^{-1}$)
representation when a single average speed approach was used. Higher emissions are emitted while
vehicles are operated with low-speed bins, which decreases the combustion efficiency of ICE and
releases more pollutants.



Figure 9b shows the road-specific breakdown between the ASD and single speed scenarios
to understand the impacts of vehicle operating speeds on onroad automobile emissions. In this
figure, each color indicates the emissions percentage differences by road types. Other than $NH_3$,
significant discrepancies happened between local urban roads (5.8%), highways (3.9%), and urban
highways (3.0%). Other pollutants, VOC, $PM_{2.5}$, CO, and $SO_x$, have similar fractions of road types.
This phenomenon is caused by low-speed conditions (<16 km h$^{-1}$) and the fractions of road VKT
contributions (Appendix C). The lower speeds cause the incomplete combustion of ICE and
increase the emission rate. Also, local urban roads, highways, and urban highways have higher
road VKT contributions at 17%, 18%, and 12%, respectively (Appendix C) than rural roads.
Higher emissions from low speed conditions from these high contribution roads (urban local, urban
highway, and highway) caused these significant differences between the ASD and single-speed
approaches. Although the interstate expressway has the largest VKT contribution (41%), it also
has the lowest fraction of low-speed bins (2%). That is why the difference between the ASD and
single speed scenarios on interstate expressways is less than 1%. In general, $NH_3$ emission factors
do not change by vehicle operating speed, so the ASD impact is quite minimal.

## 5    Conclusions

The CARS is a bottom-up automobile emissions model that utilizes the localized traffic-
related activity and emission factors input datasets to generate high quality localized bottom-up
emissions inventories for policymakers, stakeholders, and research community as well as
temporally and spatially enhanced hourly gridded emissions for CTMs. First, the CARS model
employs the daily VKTs for all registered vehicles and the emission factors function to compute
district-level total daily emissions for each vehicle. To reflect realistic traffic patterns, the CARS
model computes and utilizes link-level VKTs (=link-length×AADT) from the road network GIS
shapefiles to redistribute the original district-level total emissions into spatially enhanced road
link-level emissions. It can also optionally implement a control strategy as well as road restriction
rules to improve the quality of local emission inventories and meet the needs of users.
The CARS model is a fully modularized and computationally optimized python-based
bottom-up mobile emissions model that can effectively process a huge dataset to calculate high
quality spatiotemporal county-level, road link-level and grid cell-level mobile emissions. We
believe that the implementation of the ASD into the CARS improves the representation of onroad
automobile emissions from the road network when compared to a single-speed for each road type
approach. It allows the CARS to have a better representation of low speed (<16 km h-1) vehicle
emissions. We believe that CARS model's versatile spatiotemporal bottom-up automobile
emissions and the in-depth analysis feature can assist government policymakers and stakeholders
to develop the rapid responsive emission abatement strategies as a response to the South Korea's
national $PM_{2.5}$ emergency control strategy that enforces the nationwide vehicle restriction policy
within 24 hours.



**Code Availability:**

The source code of the CARS model public release version 1.0 can be downloaded from the
Github release website:

https://github.com/bokhaeng/CARS/releases/tag/CARSv1.0

**Digital Object Identifier (DOI) for the CARS version 1.0:**

https://zenodo.org/record/5033314#.YNzDrC1h001

**Installation Package for CARS version 1.0:**

The CARS version 1.0 installation package comes with the complete inputs and outputs datasets
for users to confirm their proper installation on their computers and can be downloaded from the
Github release website:

https://github.com/bokhaeng/CARS/releases/download/CARSv1.0/CARS_v1.0_public_release_
package_25June2021.zip

**User's Guide Documentation:**

The CARS version user's guide documentation can be accessed through the Github repository:

https://github.com/bokhaeng/CARS/tree/master/docs/User_Manual

**Data availability:**

All the datasets, excel and python scripts used in this manuscript for the data analysis are
uploaded through GMD website along with a supplemental appendix document.

**Author contribution**

Dr. B.H. Baek and Dr. Jung-Hun Woo are lead researchers in this study. Dr. Rizzieri Pedruzzi
develop the source code of CARS model, Dr. Minwoo Park tested the model and provided the
model input data. Dr. Chi-Tsan Wang analyzed the model result and prepared the manuscript.
Younha Kim, Chul-Han Song, analyzed the model result and provided comments.



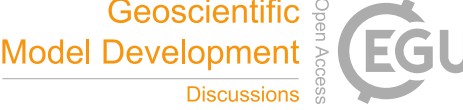

**Competing interests**

The Authors declare that they have no conflict of interest.

**Acknowledgments**

This research was funded by the National Strategic Project-Fine Particle of the National Research Foundation (NRF) of Korea funded by the Ministry of Science and ICT (MSIT), the Ministry of Environment (ME), the Ministry of Health and Welfare (MOHW) (NRF-2017M3D8A1092022), and by the Korea Environmental Industry & Technology Institute (KEITI) through the Public Technology Program based on Environmental Policy Program, funded by Korea Ministry of Environment (MOE) (2019000160007).





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



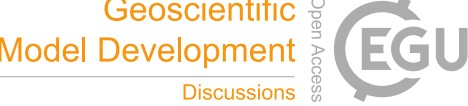

**Tables**

**Table 1**. Computational processing time by CARS module based on the modeling setup: Total number of activity data = 24,383,578; Emission Factors = 84,608; GIS road links=385,795; districts/states=5,150/16; 9km×9km grid cells=5,494 (82 columns× 67 columns).

| No | Module | Desktop i7 (minutes) | Laptop i9 (minutes) | Averaged Time (minutes) |
|----|--------|------------|-----------|---------------|
| 1 | **Process activity data** | 1.8 | 1.5 | 1.7 |
| 2 | **process emission factors** | 1.1 | 0.8 | 1.0 |
| 3 | **Process shape file** | 9.9 | 7.3 | 8.6 |
| 4 | **Calculate district emissions** | 6.4 | 5.7 | 6.1 |
| 5 | **Grid4AQM** [31days] | 4.8 [75.9] | 5.0 [87.2] | 4.9 [81.6] |
| 6 | **Plot figures** | 6.2 | 5.4 | 5.8 |
| | Total [31days] | 30.2 [101.3] | 25.7 [107.9] | 28.1[104.8] |

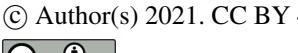



**Table 2**. The total emissions comparison between CARS and CAPSS for the 2015 emission.

| Emission Inventory | Pollutants (t yr⁻¹) | | | | | |
|---|---|---|---|---|---|---|
| | NO$_x$ | VOC | PM2.5 | CO | SO$_x$ | NH$_3$ |
| CARS 2015 | **301,794** | **61,186** | **10,108** | **373,864** | **172** | **12,453** |
| CAPSS 2015 | 369,585 | 46,145 | 8,817 | 245,516 | 209 | 10,079 |







**Table 3**. The summary tables of emissions (t yr$^{-1}$), contributions (%), and impact factor (IF, kg yr$^{-1}$)
per vehicle for criteria air pollutants (CAPs) by vehicle and fuel types: (a) for NO$_x$; (b) VOC;
(c) for PM$_{2.5}$; (d) for CO; (e) for SO$_x$; and (f) for NH$_3$.
(a) NOx

| Vehicle | Gasoline | | Diesel | | LPG | | CNG | | Hybrid | | Total | |
|---|---|---|---|---|---|---|---|---|---|---|---|---|
| | Emission | IF | Emission | IF | Emission | IF | Emission | IF | Emission | IF | Emission | IF |
| Sedan | 20,219 (6.70%) | 1.94 | 14,783 (4.90%) | 12.8 | 8,159 (2.77%) | 4.49 | 12 (0.00%) | 1.26 | 65 (0.02%) | 0.39 | 43,239 (14.3%) | 3.19 |
| Truck | 23 (0.01%) | 5.54 | **148,246 (49.1%)** | 47.9 | 920 (0.31%) | 4.55 | 88 (0.03%) | 66.4 | - | - | **149,277 (49.5%)** | 45.2 |
| Bus | 0 (0.00%) | 0.97 | 25,677 (8.51%) | **340** | - | - | 9,260 (3.07%) | 248 | 0 (0.00%) | 1.77 | 34,938 (11.6%) | **333** |
| SUV | 159 (0.05%) | 1.19 | 39,565 (13.1%) | 11.4 | 175 (0.06%) | 8.54 | 0 (0.00%) | 1.60 | 1 (0.00%) | 0.42 | 39,900 (13.2%) | 11.0 |
| Van | 14 (0.00%) | 4.78 | 16,659 (5.52%) | 22.6 | 1,337 (0.44%) | 6.80 | 0 (0.00%) | 1.25 | 0 (0.00) | 0.37 | 18,012 (6.00%) | 19.2 |
| Taxi | - | - | - | - | 1,217 (0.40%) | 2.11 | - | - | - | - | 1,217 (0.40%) | 2.11 |
| Special | 1 (0.00%) | 20.1 | 12,347 (4.10%) | 152 | 0 (0.00%) | 0.52 | - | - | - | - | 12,375 (4.10%) | 151 |
| Motorcycle | 2,836 (0.94%) | 1.31 | - | - | - | - | - | - | - | - | 2,836 (0.94%) | 1.32 |
| Total | 23,253 (7.70%) | 1.83 | **257,305 (85.3%)** | 29.9 | 11,809 (3.91%) | 4.20 | 9,361 (3.10%) | **36.7** | 66 (0.02%) | 0.39 | 301,794 (100%) | 13.3 |

(b) VOC

| Vehicle | Gasoline | | Diesel | | LPG | | CNG | | Hybrid | | Total | |
|---|---|---|---|---|---|---|---|---|---|---|---|---|
| | Emission | IF | Emission | IF | Emission | IF | Emission | IF | Emission | IF | Emission | IF |
| Sedan | 28,434 (46.5%) | 2.73 | 629 (1.03%) | 0.55 | 2,107 (3.44%) | 1.16 | 3 (0.01%) | 0.33 | 77 (0.13%) | 0.47 | **31,250 (51.1%)** | 2.30 |
| Truck | 23 (0.04%) | 5.44 | 8,194 (13.4%) | 2.65 | 286 (0.47%) | 1.41 | 102 (0.17%) | 77.2 | - | - | 8,605 (14.1%) | 2.61 |
| Bus | 0 (0.00%) | 1.65 | 717 (1.17%) | 9.51 | - | - | 11,942 (19.5%) | 320 | 0 (0.00%) | 0 | 12,659 (20.7%) | **112** |
| SUV | 246 (0.40%) | 1.84 | 2,441 (3.99%) | 0.71 | 46 (0.08%) | 2.25 | 0 (0.00%) | 0.75 | 1 (0.00%) | 0.55 | 2,733 (4.47%) | 0.76 |
| Van | 21 (0.03%) | 7.04 | 1,185 (1.94%) | 1.61 | 393 (0.64%) | 2.00 | 0 (0.00%) | 0.45 | 0 (0.00%) | 0 | 1,599 (2.61%) | 1.71 |
| Taxi | - | - | - | - | 273 (0.45%) | 0.47 | - | - | - | - | 273 (0.45%) | 0.47 |
| Special | 1 (0.00%) | 25.8 | 904 (1.48%) | 11.1 | 0 (0.00%) | 0.23 | - | - | - | - | 905 (1.48%) | 11.0 |
| Motorcycle | 3,160 (5.16%) | 1.46 | - | - | - | - | - | - | - | - | 3,160 (5.16%) | 1.46 |
| Total | **31,885 (52.1%)** | 2.50 | 14,070 (23.0%) | 1.64 | 3,106 (5.08%) | 1.10 | 12,047 (19.7%) | **247** | 78 (0.13%) | 0.47 | 61,186 (100%) | 2.51 |

(c) PM2.5

| Vehicle | Gasoline | | Diesel | | LPG | | CNG | | Hybrid | | Total | |
|---|---|---|---|---|---|---|---|---|---|---|---|---|
| | Emission | IF | Emission | IF | Emission | IF | Emission | IF | Emission | IF | Emission | IF |
| Sedan | 144 (1.42%) | 0.01 | 809 (8.00%) | 0.70 | 0 | 0 | 0 | 0 | 3 (0.03%) | 0.02 | 956 (9.46%) | 0.07 |
| Truck | 0 (0.01%) | 0 | **5,415 (53.6%)** | 1.75 | 0 | 0 | 0 | 0 | - | - | **5,415 (53.6%)** | 1.64 |
| Bus | 0 | 0 | 214 (2.11%) | **2.83** | - | - | 0 | 0 | 0 (0.01%) | 0.09 | 214 (2.11%) | 1.89 |
| SUV | 2 (0.02%) | 0.02 | 2,165 (21.4%) | 0.63 | 0 | 0 | 0 | 0 | 0 | 0 | 2,167 (21.4%) | 0.60 |
| Van | 0 | 0 | 1,127 (11.2%) | 1.53 | 0 | 0 | 0 | 0 | 0 | 0.02 | 1,127 (11.2%) | 1.20 |
| Taxi | - | - | - | - | 0 | 0 | - | - | - | - | 0 | 0 |
| Special | 0 | 0 | 230 (2.28%) | 2.82 | 0 | 0 | - | - | - | - | 230 (2.28%) | **2.81** |
| Motorcycle | 0 | 0 | - | - | - | - | - | - | - | - | 0 | 0 |
| Total | 146 (1.44%) | 0.01 | **9,959 (98.5%)** | **1.16** | 0 | 0 | 0 | 0 | 3 (0.03%) | 0.02 | 10,108 (100%) | 0.41 |






(d) CO

| Vehicle | Gasoline | | Diesel | | LPG | | CNG | | Hybrid | | Total | |
|---|---|---|---|---|---|---|---|---|---|---|---|---|
| | Emission | IF | Emission | IF | Emission | IF | Emission | IF | Emission | IF | Emission | IF |
| Sedan | 178,121 (47.6%) | 17.1 | 3,436 (0.92%) | 2.98 | 42,886 (11.5%) | 23.6 | 29 (0.01%) | 2.91 | 177 (0.05%) | 1.07 | **224,649 (60.1%)** | 16.6 |
| Truck | 254 (0.07%) | 61.1 | 47,065 (12.6%) | 15.2 | 9,088 (2.43%) | 44.9 | 68 (0.02%) | 51.4 | - | - | 56,475 (15.1%) | 17.1 |
| Bus | 0 (0.00%) | 19.3 | 7,633 (2.05%) | **101** | - | - | 1542 (0.41%) | 41.3 | 1 (0.00%) | 4.64 | 9,176 (2.45%) | **81.2** |
| SUV | 2,616 (0.70%) | 19.6 | 13,401 (3.58%) | 3.87 | 791 (0.21%) | 38.6 | 0 (0.00%) | 4.09 | 2 (0.00%) | 1.15 | 16,808 (4.50%) | 4.65 |
| Van | 131 (0.04%) | 43.4 | 6,611 (1.77%) | 8.97 | 8,032 (2.15%) | 40.9 | 2 (0.00%) | 6.53 | 0 (0.00%) | 1.00 | 14,777 (3.95%) | 15.8 |
| Taxi | - | - | - | - | 8,481 (2.27%) | 14.7 | - | - | - | - | 8,481 (2.27%) | 14.7 |
| Special | 13 (0.00%) | 269 | 4,224 (1.13%) | 51.7 | 1 (0.00%) | 3.69 | - | - | - | - | 4,239 (1.13%) | 51.7 |
| Motorcycle | 39,256 (10.5%) | 18.2 | - | - | - | - | - | - | - | - | 39,256 (10.5%) | 18.2 |
| Total | **220,390 (59.0%)** | 17.3 | 82,372 (22.0%) | 9.57 | 69,281 (18.5%) | **24.6** | 1641 (0.44%) | 33.6 | 180 (0.05%) | 1.07 | 373,864 (100%) | 15.4 |

(e) SO$_x$

| Vehicle | Gasoline | | Diesel | | LPG | | CNG | | Hybrid | | Total | |
|---|---|---|---|---|---|---|---|---|---|---|---|---|
| | Emission | IF | Emission | IF | Emission | IF | Emission | IF | Emission | IF | Emission | IF |
| Sedan | 51.3 (29.8%) | 0.005 | 6.5 (3.79%) | 0.006 | 8.28 (4.81%) | 0.005 | 0 | 0 | 1.14 (0.67%) | 0.007 | **67.2 (39.1%)** | 0.005 |
| Truck | 0.03 (0.02%) | 0.008 | 45.5 (26.5%) | 0.015 | 0.97 (0.57%) | 0.005 | 0 | 0 | - | - | 46.5 (27.1%) | 0.014 |
| Bus | 0 (0.00%) | 0.003 | 10.8 (6.26%) | **0.143** | - | - | 0 | 0 | 0.01 (0.01%) | 0.047 | 10.8 (6.26%) | **0.095** |
| SUV | 0 (0.00%) | 0.000 | 18.2 (10.6%) | 0.005 | 0.00 (0.00%) | 0.000 | 0 | 0 | 0.01 (0.01%) | 0.007 | 18.2 (10.6%) | 0.005 |
| Van | 0.02 (0.01%) | 0.006 | 5.5 (3.20%) | 0.007 | 0.77 (0.45%) | 0.004 | 0 | 0 | 0 (0.00%) | 0.010 | 6.30 (3.66%) | 0.007 |
| Taxi | - | - | - | - | 7.71 (4.49%) | 0.013 | - | - | - | - | 7.71 (4.48%) | 0.013 |
| Special | 0 (0.00%) | 0.003 | 7.3 (4.27%) | 0.090 | 0.00 (0.00%) | 0.005 | - | - | - | - | 7.34 (4.27%) | 0.090 |
| Motorcycle | 7.94 (4.62%) | 0.004 | - | - | - | - | - | - | - | - | 7.94 (4.62%) | 0.004 |
| Total | 59.3 (34.5%) | 0.006 | **93.8 (54.5%)** | **0.011** | 17.7 (10.3%) | 0.006 | 0 | 0 | 1.17 (0.68%) | 0.007 | 172 (100%) | 0.007 |

(e) NH$_3$

| Vehicle | Gasoline | | Diesel | | LPG | | CNG | | Hybrid | | Total | |
|---|---|---|---|---|---|---|---|---|---|---|---|---|
| | Emission | IF | Emission | IF | Emission | IF | Emission | IF | Emission | IF | Emission | IF |
| Sedan | 12,225 (98.3%) | **1.17** | 20 (0.16%) | 0.02 | 0 | 0.00 | 0 | 0 | 19 (0.15%) | 0.11 | **12,284 (98.6%)** | **0.91** |
| Truck | 0 (0.00%) | 0.03 | 82 (0.66%) | 0.03 | 0 | 0.00 | 0 | 0 | - | - | 82 (0.66%) | 0.02 |
| Bus | 0 (0.00%) | 0.09 | 15 (0.12%) | 0.19 | - | - | 0 | 0 | 0 (0.00%) | 0.51 | 15 (0.12%) | 0.13 |
| SUV | 0 (0.00%) | 0.00 | 0 (0.00%) | 0.00 | 0 | 0.00 | 0 | 0 | 0 (0.00%) | 0.16 | 0 (0.00%) | 0.00 |
| Van | 0 (0.00%) | 0.02 | 14 (0.11%) | 0.02 | 0 | 0.00 | 0 | 0 | 0 (0.00%) | 0.09 | 14 (0.11%) | 0.01 |
| Taxi | - | - | - | - | 0 | 0.00 | - | - | - | - | 0 (0.00%) | 0.00 |
| Special | 0 (0.00%) | 0.01 | 10 (0.08%) | 0.12 | 0 | 0.00 | - | - | - | - | 10 (0.08%) | 0.12 |
| Motorcycle | 49 (0.39%) | 0.02 | - | - | - | - | - | - | - | - | 49 (0.39%) | 0.02 |
| Total | **12,293 (98.7%)** | **0.97** | 141 (1.13%) | 0.02 | 0 | 0.00 | 0 | 0 | 19 (0.16%) | 0.12 | 12,453 (100%) | 0.51 |




**Figures**

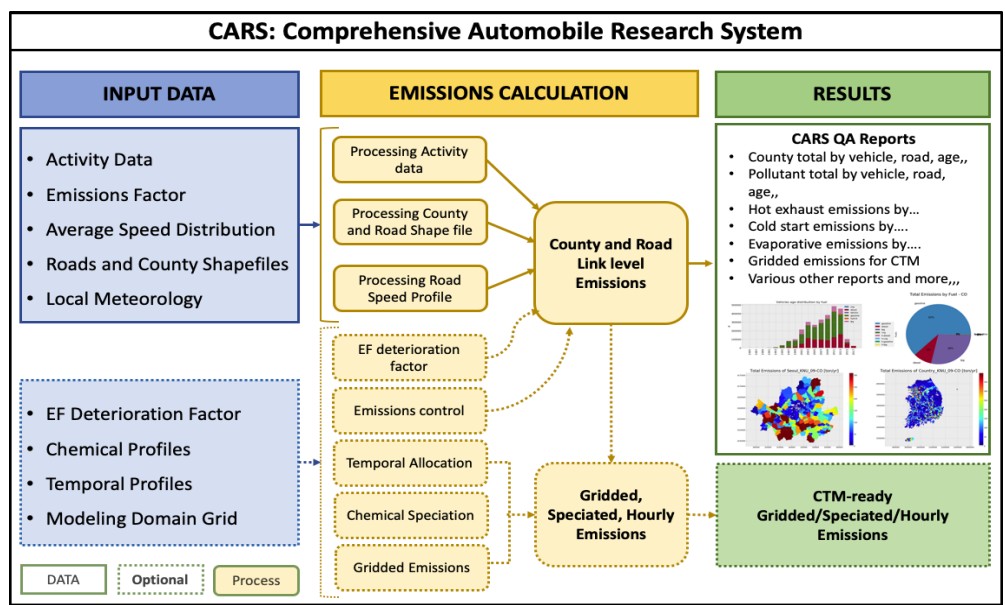

**Figure 1**. CARS schematic methodology to estimate mobile emissions.





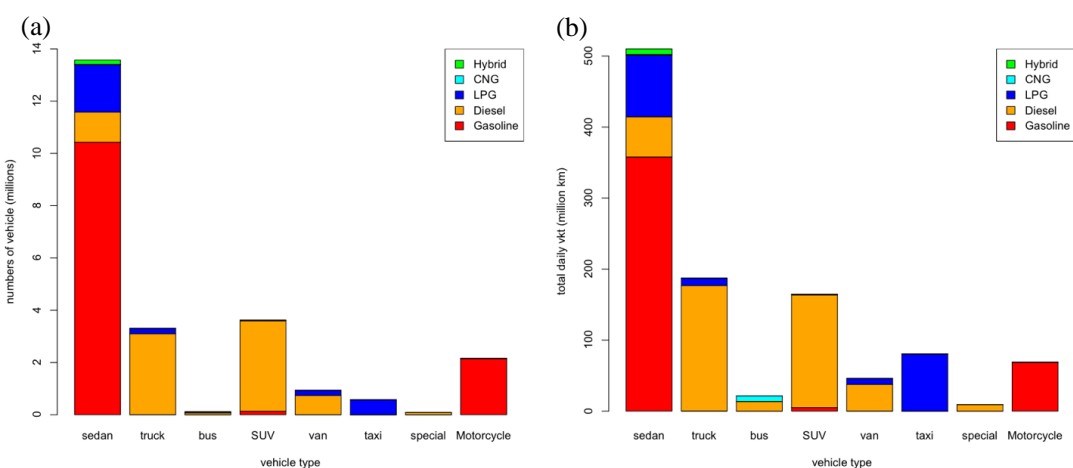


**Figure 2. (a)** The number of vehicles by vehicle and fuel types and **(b)** the total daily VKT by
vehicle and fuel types in South Korea.



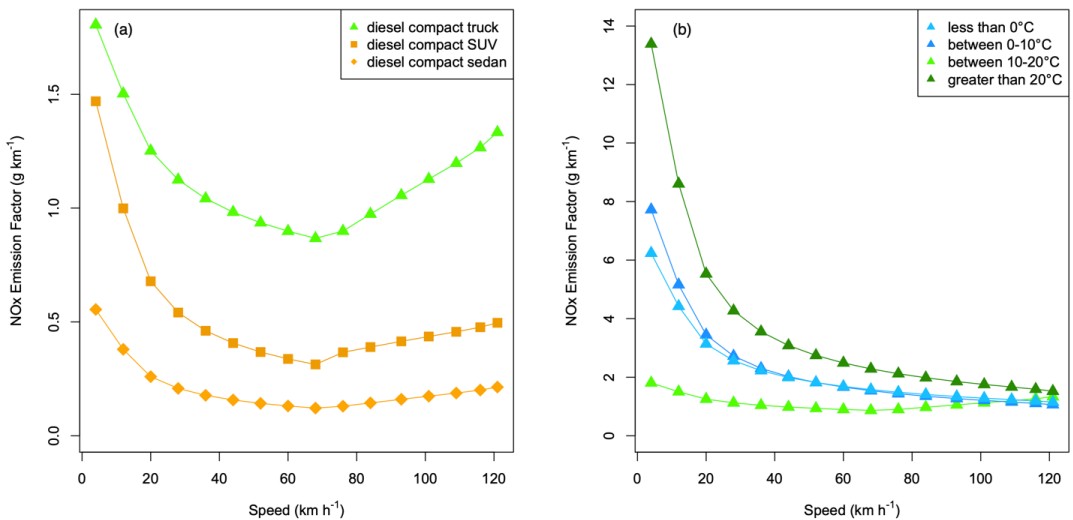


**Figure 3**. Variation of NOx emission factors from diesel compact engines by vehicle speed and ambient temperatures: **(a)** $NO_x$ emission factors function to vehicle speed; **(b)** $NO_x$ emission factors of diesel compact truck function to vehicle speed and ambient temperature.


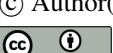


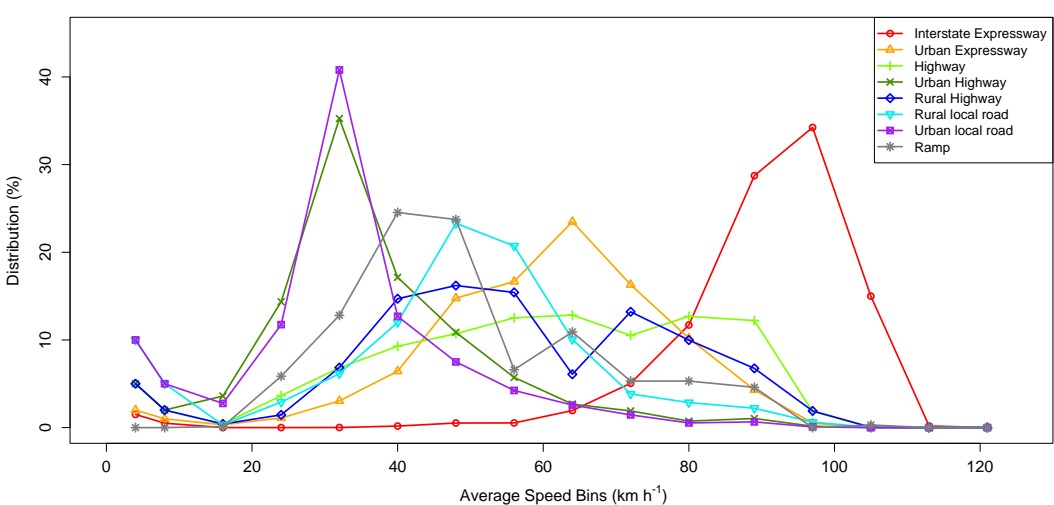

**Figure 4**. Road-specific average speed distribution (ASD) in South Korea.





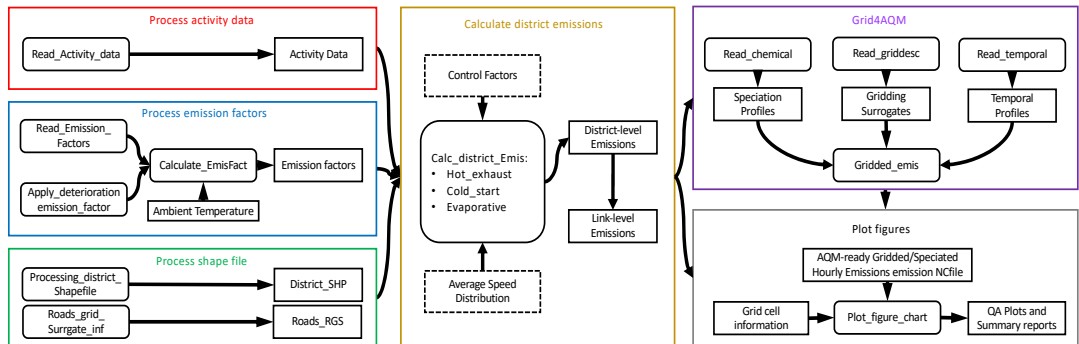

**Figure 5**. The schematic of modules and their functions in the CARS.




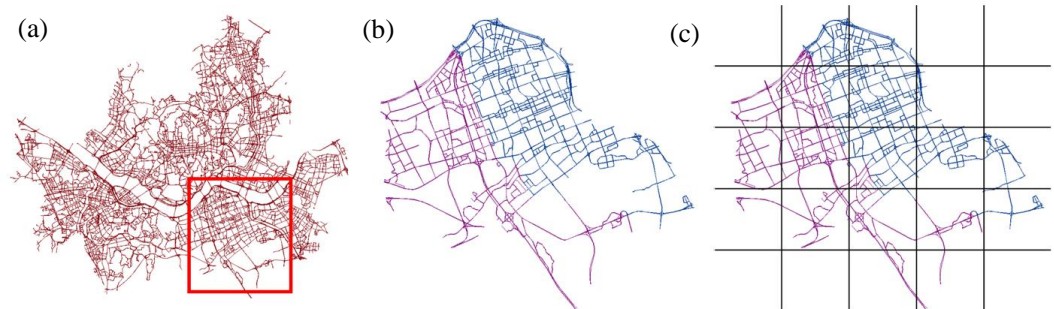

**Figure 6 (a)** the road network GIS shapefile of Seoul, South Korea; **(b)** two districts with different
colors (purple and blue); **(c)** the modeling grid cells over road segments.



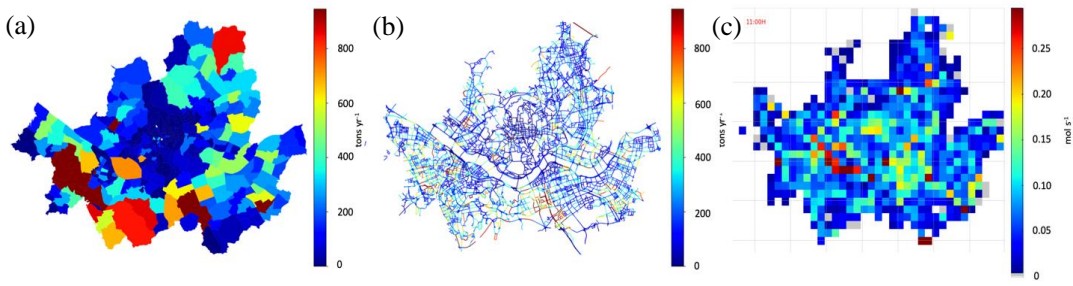


**Figure 7**. Three different formats of CO emissions from CARS, (A) District-level total emissions
(t yr$^{-1}$) (B) Link-level total emissions (t yr$^{-1}$), (C) CTM-ready gridded hourly total emissions (moles
s$^{-1}$).


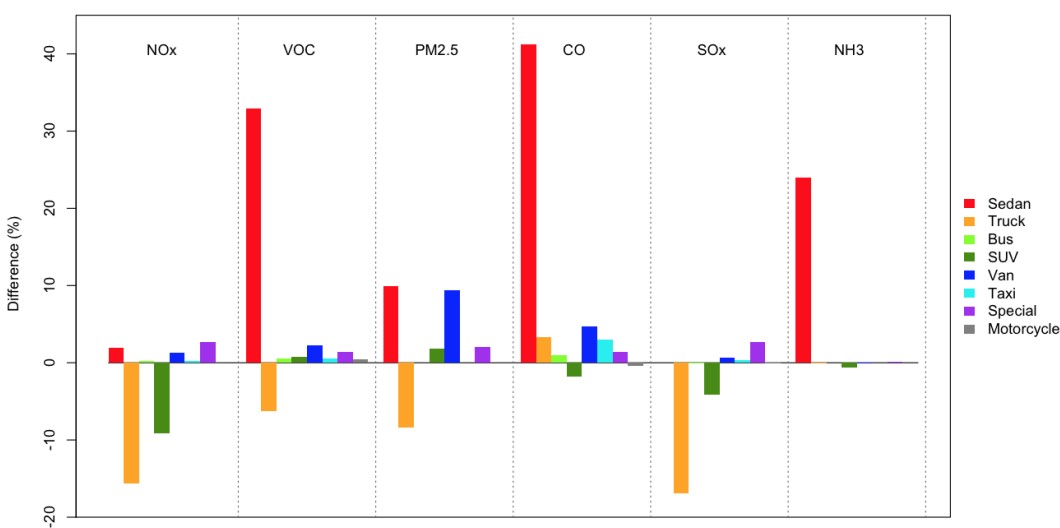

**Figure 8**. Comparison between CARS 2015 and CAPSS 2015 onroad mobile emissions
inventories by vehicle types. The standard line is CAPSS 2015 data.






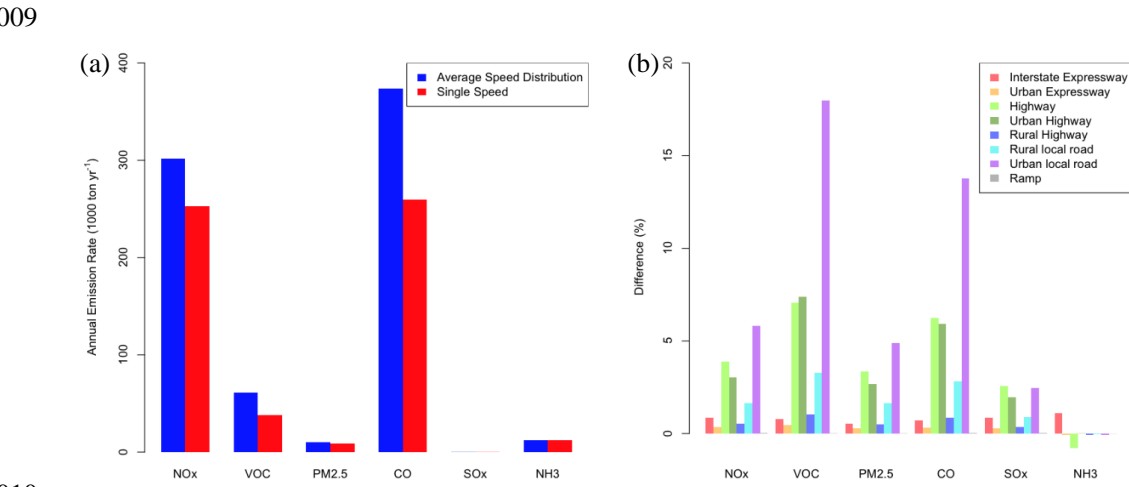


**Figure 9**. The impacts of emissions between the ASD and single-speed approach: (a) the total
emission differences by pollutant; (b) The road-specific difference (%) by pollutant.



**Appendics**

**Appendix A**: The vehicle types classified by fuel type, vehicle body type, and engine size. The
emission factors of the diesel vehicle with the star (*) are depended on the ambient temperature
(*T*).

| Vehicle Types | Fuel Types | | | | | | | |
|---|---|---|---|---|---|---|---|---|
| | Gasoline | Diesel | LPG | CNG | HYBRID_G | HYBRID_D | HYBRID_L | HYBRID_C |
| Sedan | Supercompact | Supercompact* | Supercompact | - | - | - | - | - |
| | Compact | compact* | compact | compact | compact | compact | compact | - |
| | Fullsize | Fullsize* | Fullsize | Fullsize | Fullsize | Fullsize | Fullsize | - |
| | Midsize | Midsize* | Midsize | Midsize | Midsize | Midsize | Midsize | - |
| Truck | Supercompact | Supercompact | Supercompact | - | - | - | - | - |
| | Compact | Compact* | Compact | Compact | - | - | - | - |
| | Fullsize | Concrete | - | Fullsize | - | - | - | - |
| | Midsize | Fullsize | Midsize | Midsize | - | - | - | - |
| | - | Midsize | - | - | - | - | - | - |
| | - | Dump | - | - | - | - | - | - |
| | - | Special | Special | Special | - | - | - | - |
| Bus | Urban | Urban | Urban | Urban | - | Urban | - | - |
| | - | Rural | - | Rural | - | Rural | - | Rural |
| SUV | Compact | Compact* | Compact | - | - | - | - | - |
| | Midsize | Midsize* | Midsize | Midsize | Midsize | - | - | - |
| Van | supercompact | supercompact | supercompact | - | - | - | - | - |
| | Compact | Compact | Compact | Compact | - | - | - | - |
| | - | - | Fullsize | Fullsize | Fullsize | Fullsize | Fullsize | Fullsize |
| | Midsize | Midsize | Midsize | Midsize | Midsize | Midsize | Midsize | Midsize |
| Taxi | - | - | Compact | - | - | - | - | - |
| | - | - | Fullsize | - | - | - | - | - |
| | - | - | Midsize | - | - | - | - | - |
| Special | - | Tow | - | - | - | - | - | - |
| | Wrecking | Wrecking | Wrecking | Wrecking | - | - | - | - |
| | Others | Others | Others | - | - | - | - | - |
| Motorcycle | Compact | - | - | - | - | - | - | - |
| | Midsize | - | - | - | - | - | - | - |
| | Fullsize | - | - | - | - | - | - | - |

- no existence
* ambient temperature-dependent diesel vehicle
LPG: Liquefied Petroleum Gas
CNG: Connecticut Natural Gas
Hybrid_G: hybrid vehicle with gasoline
Hybrid_D: hybrid vehicle with diesel
Hybrid_L: hybrid vehicle with LPG
Hybrid_C: hybrid vehicle with CNG






**Appendix B**, The summary of activity data (number of vehicles and daily total VKTs) in South Korea by vehicle type with engine size.

| Vehicle Types | Engine sizes | Fuel Types | | | | | | | | | |
|---|---|---|---|---|---|---|---|---|---|---|---|
| | | Gasoline | | Diesel | | LPG | | CNG | | Hybrid | |
| | | Numbers | Daily VKT | Numbers | Daily VKT | Numbers | Daily VKT | Numbers | Daily VKT | Numbers | Daily VKT |
| Sedan | Supercompact | 1,792,471 | 50,197,345 | 46 | 1,761 | 83,226 | 4,000,067 | 6 | 237 | - | - |
| | Compact | 1,372,317 | 39,543,668 | 51,324 | 2,570,086 | 8,040 | 257,060 | 276 | 12,115 | 3,802 | 137,360 |
| | Fullsize | 2,403,327 | 100,632,702 | 428,831 | 20,928,552 | 292,850 | 15,910,588 | 5,296 | 323,852 | 21,533 | 1,086,509 |
| | Midsize | 4,858,533 | 167,454,032 | 672,960 | 33,126,318 | 1,431,970 | 66,640,378 | 4,310 | 625,717 | 140,527 | 6,717,856 |
| Truck | Supercompact | 850 | 9,595 | 816 | 354 | 111,051 | 6,550,476 | - | - | - | - |
| | Compact | 3,185 | 143,510 | 2,655,089 | 133,480,216 | 87,650 | 3,567,109 | 42 | 2,694 | - | - |
| | Fullsize | 3 | 422 | 180,991 | 25,774,819 | - | - | 72 | 4,676 | - | - |
| | Midsize | 98 | 7,430 | 258,509 | 17,477,685 | 1,434 | 47,870 | 14 | 483 | - | - |
| | Dump | - | - | - | - | - | - | - | - | - | - |
| | Special | 20 | 970 | - | - | 2,292 | 99,124 | 1,194 | 60,886 | - | - |
| Bus | Urban | 1 | 126 | 40,448 | 7,282,593 | 1 | 652 | 6,543 | 1,466,854 | 2 | 282 |
| | Rural | - | - | 34,997 | 6,334,278 | - | - | 30,792 | 6,460,001 | 216 | 50,873 |
| SUV | Compact | 42,348 | 1,395,153 | 2,341,397 | 105,962,626 | 6,946 | 275,728 | 13 | 551 | - | - |
| | Midsize | 91,002 | 3,520,552 | 1,120,128 | 5,277,861 | 13,567 | 595,426 | 15 | 706 | 1,719 | 88,683 |
| Van | supercompact | 88 | 1,645 | - | - | 44,947 | 2,058,014 | - | - | - | - |
| | Compact | 2,937 | 87,507 | 685,317 | 34,781,937 | 151,654 | 6,135,138 | 7 | 255 | - | - |
| | Fullsize | - | - | 19,452 | 1,318,221 | 1 | 14 | 97 | 7,598 | 3 | 136 |
| | Midsize | 2 | 1,303,795 | 31,790 | 1,433,407 | 15 | 416 | 160 | 15,216 | 2 | 85 |
| | Special | - | - | - | - | - | - | - | - | - | - |
| Taxi | Compact | - | - | - | - | 8,380 | 576,378 | - | - | - | - |
| | Fullsize | - | - | - | - | 92,861 | 10,827,756 | - | - | - | - |
| | Midsize | - | - | - | - | 474,455 | 69,087,721 | - | - | - | - |
| Special | Tow | - | - | 40,807 | 7,447,773 | - | - | - | - | - | - |
| | Wrecking | 2 | 138 | 12,568 | 813,746 | 128 | 6,607 | 3 | 94 | - | - |
| | Others | 47 | 553 | 28,275 | 989,988 | 180 | 9,966 | - | - | - | - |
| Motorcycle | Compact | 184,822 | 3,507,948 | - | - | - | - | - | - | - | - |
| | Fullsize | 65,964 | 3,493,728 | - | - | - | - | - | - | - | - |
| | Midsize | 1,910,988 | 61,676,824 | - | - | - | - | - | - | - | - |

- no existence

LPG: Liquefied Petroleum Gas

CNG: Connecticut Natural Gas

Hybrid: all hybrid vehicles, electric power mixed with fossil fuel (gasoline, diesel, LPG, or CNG)






**Appendix C**, Eight road types with assigned average vehicle operating speed and VKT fractions.

| Road types | Description | Average Speed (km h$^{-1}$) | Road VKT fraction |
|---|---|---|---|
| 101 | Interstate Expressway | 90 | 41% |
| 102 | Urban Expressway | 60 | 5% |
| 103 | Highway | 58 | 18% |
| 104 | Urban Highway | 36 | 12% |
| 105 | Rural Highway | 55 | 3% |
| 106 | Rural Local Road | 45 | 4% |
| 107 | Urban Local Road | 32 | 17% |
| 108 | Ramp | 50 | 0.4% |


**Appendix D,** The daily average VKT (km d$^{-1}$) per vehicle by vehicle and fuel types.

| Vehicle types | Fuel Types | | | | | |
|---|---|---|---|---|---|---|
| | Gasoline | Diesel | LPG | CNG | Hybrid | Average |
| Sedan | 34 | 49 | 48 | 97 | 48 | 38 |
| Truck | 39 | 57 | 51 | 52 | - | 57 |
| Bus | 126 | 180 | - | 212 | 237 | 191 |
| SUV | 37 | 46 | 42 | 45 | 52 | 46 |
| VAN | 29 | 51 | 42 | 87 | 44 | 49 |
| Taxi | - | - | 140 | - | - | 140 |
| Special | 14 | 113 | 54 | 31 | - | 113 |
| Motorcycle | 32 | - | - | - | - | 32 |







**Appendix E**, Average speed distribution (ASD) for each road type: The table columns are
different road types, and the table rows are average speed of each speed bin.

| Speed (km/d) | Road Types | | | | | | | |
|---|---|---|---|---|---|---|---|---|
| | 101 | 102 | 103 | 104 | 105 | 106 | 107 | 108 |
| 4 | 1.50% | 2.00% | 5.00% | 5.00% | 5.00% | 10.00% | 10.00% | 0.00% |
| 8 | 0.50% | 1.00% | 2.00% | 2.00% | 2.00% | 5.00% | 5.00% | 0.00% |
| 16 | 0.00% | 0.33% | 0.40% | 3.59% | 0.41% | 0.30% | 2.76% | 0.11% |
| 24 | 0.00% | 1.09% | 3.64% | 14.35% | 1.45% | 2.91% | 11.75% | 5.85% |
| 32 | 0.01% | 3.04% | 6.82% | 35.25% | 6.85% | 6.15% | 40.80% | 12.80% |
| 40 | 0.17% | 6.43% | 9.28% | 17.14% | 14.70% | 12.00% | 12.69% | 24.53% |
| 48 | 0.52% | 14.76% | 10.70% | 10.86% | 16.20% | 23.30% | 7.49% | 23.74% |
| 56 | 0.53% | 16.66% | 12.52% | 5.72% | 15.42% | 20.72% | 4.24% | 6.60% |
| 64 | 1.94% | 23.49% | 12.83% | 2.68% | 6.08% | 10.06% | 2.56% | 10.90% |
| 72 | 5.05% | 16.30% | 10.51% | 1.90% | 13.21% | 3.84% | 1.45% | 5.30% |
| 80 | 11.70% | 10.19% | 12.69% | 0.74% | 9.98% | 2.85% | 0.53% | 5.30% |
| 89 | 28.73% | 4.30% | 12.21% | 1.04% | 6.75% | 2.21% | 0.65% | 4.59% |
| 97 | 34.24% | 0.51% | 1.82% | 0.15% | 1.90% | 0.62% | 0.08% | 0.00% |
| 105 | 14.99% | 0.00% | 0.02% | 0.00% | 0.04% | 0.03% | 0.00% | 0.30% |
| 113 | 0.18% | 0.00% | 0.00% | 0.00% | 0.00% | 0.00% | 0.00% | 0.00% |
| 121 | 0.01% | 0.00% | 0.00% | 0.00% | 0.00% | 0.00% | 0.00% | 0.00% |

**Appendix F**: A single-speed for each road type

| Speed (km/d) | Road Types | | | | | | | |
|---|---|---|---|---|---|---|---|---|
| | 101 | 102 | 103 | 104 | 105 | 106 | 107 | 108 |
| 4 | 0% | 0% | 0% | 0% | 0% | 0% | 0% | 0% |
| 8 | 0% | 0% | 0% | 0% | 0% | 0% | 0% | 0% |
| 16 | 0% | 0% | 0% | 0% | 0% | 0% | 0% | 0% |
| 24 | 0% | 0% | 0% | 0% | 0% | 0% | 0% | 0% |
| 32 | 0% | 0% | 0% | 0% | 0% | 0% | 100% | 0% |
| 40 | 0% | 0% | 0% | 100% | 0% | 0% | 0% | 0% |
| 48 | 0% | 0% | 0% | 0% | 0% | 100% | 0% | 100% |
| 56 | 0% | 0% | 100% | 0% | 100% | 0% | 0% | 0% |
| 64 | 0% | 100% | 0% | 0% | 0% | 0% | 0% | 0% |
| 72 | 0% | 0% | 0% | 0% | 0% | 0% | 0% | 0% |
| 80 | 0% | 0% | 0% | 0% | 0% | 0% | 0% | 0% |
| 89 | 100% | 0% | 0% | 0% | 0% | 0% | 0% | 0% |
| 97 | 0% | 0% | 0% | 0% | 0% | 0% | 0% | 0% |
| 105 | 0% | 0% | 0% | 0% | 0% | 0% | 0% | 0% |
| 113 | 0% | 0% | 0% | 0% | 0% | 0% | 0% | 0% |
| 121 | 0% | 0% | 0% | 0% | 0% | 0% | 0% | 0% |
