# Peer review of "Comprehensive Automobile Research System (CARS) – a"

_Geoscientific Model Development, 2021_

## Referee Comment (RC2)

**General Comments**

Development of a new open-source vehicle emissions model has merit. However, I am concerned about the rationale in using the average speed distribution from the State of Georgia in use in this study, also with the language suggesting that it is an improvement in accuracy. I am concerned there is a disconnect in the use of average speed distribution in MOVES and in the CARS model, and don't understand why an average speed distribution could be calculated from the link-level data from South Korea's GSI road shape files. I added substantial questions and comments regarding this issue in the specific comments below. (See Page 8. Line 271-281).

Because I felt this issue must be addressed before going forward with the paper, I stopped my review at this point. And only conducted a cursory review of the remaining aspects of the paper, including results and conclusion.

I have also added many specific comments to remove generalizations that may be not accurate, provide citations behind some of the statements, and to clarify the calculations which I encountered in the abstract, introduction and methods section.

I am willing to re-review the paper if my concerns can be addressed regarding the average speed distributions in the model.

**Specific comments (***suggested text in italics***)**

p.1 Line 24 "it can optionally utilize road link-specific average speed distribution (ASD)"

 - is it an average speed distribution of the road type, or the individual link? The wording is not clear.

Should it be referred to road-specific average speed distribution? Like on p.1 line 32?

p.2 Line 39-41 "It indicates that the CNG bus is better for the rural area while the diesel bus is better applicable for the urban area for a better ozone control strategy because the rural area is usually NOx limited for ozone formation and urban area is VOC limited region"

Is this backed up with air quality modeling, or assumed based on the reasons given here? In practice, couldn't it be much more complex? If it is not built on analysis, then I think the statement should be re-written as a potential ozone control strategy—which may need to be backed up with more analysis. E.g. what would be the impact on suburban areas of more NOx or VOC emissions?

p.2. line 47. The line about indoor vs. total air pollution makes it seem that ambient air pollution is a relatively minor contributor to public health. I would add more citations to clarify that ambient pollution impacts indoor air quality, or remove the indoor air quality references—as potentially misleading.

For example: Cohen et al. 2017 estimate 4.2 annual early deaths to ambient PM.

Cohen, A. J., et al. (2017). Estimates and 25-year trends of the global burden of disease attributable to ambient air pollution: an analysis of data from the Global Burden of Diseases Study 2015. *The Lancet,* 389 (10082), 1907-1918. DOI: https://doi.org/10.1016/S0140-6736(17)30505-6.

Burnett, et a. 2018 estimate the health burden is closer to 9 million deaths from ambient PM concentrations

Burnett, R., et al. (2018). Global estimates of mortality associated with long-term exposure to outdoor fine particulate matter. Proceedings of the National Academy of Sciences, 115 (38), 9592-9597. DOI: 10.1073/pnas.1803222115.

p. 2. Line 57-59. Is this statement backed up with a citation? If not, I would not say this could be an over-generalized statement is generally always accurate. For example, in areas with persistent cold pooling inversions—modeling the meteorology and chemistry may just or more critical then accurately modeling the emissions—correct?

p.2 60-61. Another over-generalized statement that deserves more context and a citation. For which pollutant? Do you mean for NOx? I don't think this is true for VOCs, and was a little surprised that it was mentioned as true for PM2.5 (because that is not the case with the US NEI for select urban counties).

p. 2 lines 67-69. Are these studies based on air quality modeling or observations? Either way, I would suggest these results are presented in the context of other studies that show that primary PM2.5 is a minor contributor to ambient PM2.5.

Nault et al. 2021 suggests that PM2.5 in Seoul (and other major urban areas across the world) have only a small contribution of PM2.5 from primary PM2.5. Most is secondary formed PM either as secondary organic aerosol or secondary inorganic PM2.5 (ammonium-sulfate or ammonium-nitrate).

> Nault, B. A., et al. (2021). Secondary organic aerosols from anthropogenic volatile organic compounds contribute substantially to air pollution mortality. Atmos. Chem. Phys., 21 (14), 11201-11224. DOI: 10.5194/acp-21-11201-2021.
>
> See also
>
> Jimenez, J. L., et al. (2009). Evolution of Organic Aerosols in the Atmosphere. Science, 326 (5959), 1525-1529. DOI: 10.1126/science.1180353.

p. 3. Line 73. " *highly resolved* spatiotemporal automobile emissions"

- bottom-up emissions inventories by process can give high resolution, but are not necessarily higher quality than top-down methods.

Aren't the vehicle operation processes both physical and chemical? I would recommend stating that you can get more spatiotemporally resolved emissions inventories when pairing process-specific emissions models with resolved vehicle activity data.

p. 3. Line 97-98. They are developed differently to meet *specific user needs?*  based on the types of traffic

activity and emission factors, emission calculation methodologies, and other optional/available.

- I hope the emissions models are not developed to meet the models own needs....but on the model users needs
- Should you mention that each model is developed with different levels of specificity, underlying data sets, and modeling assumptions?

p.3. lines 100-101. This statement is not clear, and not sure it is needed here.

P. 108-109. I disagree about the general statement on the lack of transparency for emission factors. Technical reports that document the emission factors and algorithms for estimating emissions are available here: https://www.epa.gov/moves/moves-onroad-technical-reports

Perhaps, it could be stated that it is high degree of specificity, make it difficult to update and apply to countries outside the USA. (although there are examples of the done in the literature).

Page 6. Line 180-181. What do you mean by traffic density? Are you referring to total VKT as the subtitle suggests? Isn't VKT a measure of traffic flow rather than density?

Page 6. Line 192-193. How is the VIN used in the calculation? Is that to calculate the vehicle age? Could you clarify?

p. 6 line 194-195. VKT with the manufactured year (VKTv,age) is calculated based on the cumulative mileage (Mf)  *between* the last inspection date (Df) and registration date (D0).

  - Clarification on the data in the calculation. Is the registration date always at or near age zero? Does the registration data only capture vehicles when they change ownership? Or does that happen more regularly? I think it is important to clarify if the VKT calculated in equation 1, is reflective of most recent years of use, or an average VKT over the lifetime of the vehicle.

  - Also, since equation 1 is applied to individual vehicles, shouldn't the subscript reflect individual vehicle in the equations, rather than just vehicle type, and age?

p. 6 lines 205-207. 'nonroad automobile' is potentially confusing. Vehicles in MOVES are classified as onroad or nonroad. When passenger vehicles are operating in driveways, parking lots, I would classify this as off-network automobile emissions.

  Can you clarify what are the off-network automobile emissions that are missing? Are those starts? Evaporative? Idling emissions? Or are they spatially allocated to the roadways.

Line 225-227. Should you also mention that emissions are produced from incomplete combustion products that are not controlled from the emissions aftertreatment equipment, such as a three-way catalytic converter for gasoline vehicles?

Line 228- I would not say NOx is similarly produced as SOx, because the source of S is the fuel, not the atmosphere like nitrogen.

Page 7. Line 280-235. Does the age in the emission factor, reflect the impact of model year on the new vehicle emission rate, and age in the DF factor reflect aging effects from the new vehicle emission rate? If so, could that be clarified—otherwise, it is not clear what the purpose of DF is. However, if everything is calculated in terms of age—does that mean an emission rate for a 5 year-old vehicle is the same, for all calendar years? Or does model only work for one calendar year?

Page 8. 247-248. In this equation, is vehicle age used to reflect the model year or technology effect of a new vehicle, or a deterioration effect? If everything is in terms

Page 8. 255 "Figure 3a shows a significant decrease of NOx emissions while speed increases *between 0 and 70 kmh*.

Page 8. Line 260. I would remove word 'constant' speed. That implies that the vehicle emissions model can differentiate between emission factors between constant speeds, and transient sec/sec speeds, which I do not think is the case. My understanding is that you are differentiating between a single average speed, to an average speed distribution. However, the associated emission rates with the average speeds is not changing.

Page 8. Line 262. Remove 'incomplete ICE combustion' higher NOx emissions are not necessary directly related to an issue with incomplete combustion—NOx emissions can be lower when ICE conditions are fuel rich.  I would just recommend that a single speed, may not represent the average emission rates, as an average speed distribution with time spend at multiple speeds.

Page 8. Line 271-281.

I believe there may be an issue with using ASD from MOVES for Korea in CARS.

1. Average speed distributions from MOVES and provided from the State of Georgia should be defined. MOVES average speed distributions are intended to calculate the distribution of average speeds across multiple links within the same road type. As well, as to capture the distribution in average speeds in links across time (such as days with traffic incidents or normal travel days). In practice, the data is aggregated from telematics data that calculate average speeds from varying resolutions (1 hz to every 180 seconds). As documented in the MOVES population and activity report.

   USEPA (2020). Population and Activity of Onroad Vehicles in MOVES3.   EPA-420-R-20-023. Office of Transportation and Air Quality. US Environmental Protection Agency. Ann Arbor, MI. November 2020. https://www.epa.gov/moves/moves-technical-reports.

   I am not familiar with how the State of Georgia average speed distributions were calculated, but that should be explained if those are used in the study.

2. Why not develop average speed distributions from South Korea's GSI road shape files? The text seems to suggest that it is a problem that there is only one average speed associated with each road link. But this is not a problem. There should be many links of the same road type within a region. Using that data, you can calculate an average speed distribution for that road type in the region. Is it because you want to capture variation in link levels average speeds across different days or hours of the day? If so please clarify.

3. It is not mentioned if the average speed distributions vary by time of day—please clarify, they do in MOVES, which captures the effect of the diurnal traffic pattern on vehicle speeds.

4. The emission rates from the CARS model—are they intended to be associated with a cycle average speed that represents the average speed with a road link (which contains variation of speed within the link)? Or are the emission rates intended to be associated with sec/sec speed data? If the emission rates are intended to be cycle or link average emission rates, then it makes sense to use average speed distribution calculated from the average speeds from many links, rather than using an average speed distribution calculated from sec/sec data.

5. Is using the average speed distribution approach even needed here? If you have the average speed for each individual road link? Why not use that approach? Add an explanation.

6. It is not clear why using ASD from the state of Georgia are more realistic than the using the current data. Also, the explanation is not clear on the development of the inputs. E.g. where did the 2:1 ratio of bins 1 and 2 come from. And where did the additional 2%, 3%, 7%, and 15% come from. Was it added such that the average speeds on each roadway is the same before and after the fix?
7. To me, it seems like this should be better classified as a 'sensitivity' study on the potential impact of using average speed distributions. Since, the data for the average speed distributions used in the study, don't seem to be clearly better than what is there, and seem to be based on quite a few assumptions.

**Technical corrections: (**_suggested text in italics)_

p. 1. Line 18. "utilize  local vehicle activity data"

p.1 Line 20 "to generate a temporally and spatially _resolved_ "

p. 2 line 28, "due to  _it having the_ longest daily VKT _and relatively high NOx g/km emission rate."_

p. 4. Line 133 "changing specific variables that may be  _embedded in the code?"_

p.5 line 151 "road link-level"

p.5 line 174. "South Korean traffic databases _from_ "

line 224 "hot exhaust emission_s"_

p. 46 Appendix E and F. Speed units should be km/hour. Also, what is the range of the speed bins? That is not clear from the Tables.

p. Appendix F. Single _average_ speed for each road type

The grammar should be improved. I recommend that it be reviewed by an English technical editor. I started marking grammatical corrections on the abstract but did not make comments on the rest of the document.

---

## Author Response (AR1)

**Responsed to the Reviewers**

We would like to thank the reviewers for their comments and believe that it has improved the manuscript. The reviewers' comments are in grey italics and our response is given in black.

**Referee #1 Comments**

*General Comment*
*This manuscript describes the CARSv1 system, a python-based automobile emissions inventory model that allows estimating high-resolution emissions from road transport activities. The strength of CARS is in its ability and flexibility to generate emission results in multiple formats and for multiple purposes, ranging from policymaking to air quality modelling. Moreover, the system makes use of very detailed and local input datasets, which allows computing emissions with a high level of representativeness. Emissions computed by CARS for South Korea are presented and compared against a local emission inventory to illustrate its capabilities and to show the high sensitivity of the results to the vehicle operating speed. The paper is very well written and structured, and its quality is excellent, which makes it a very good contribution to GMD. I therefore recommend to accept this manuscript for publication once the following minor comments have been addressed.*

*CARS estimates hot exhaust, cold start, and evaporative emissions from road transport. However, PM emissions from non-exhaust processes (i.e., tyre, road and brake wear,resuspension) are not included in the calculation process. Several studies have highlighted that non-exhaust PM emissions can dominate total traffic PM10 emissions (e.g., Denier van der Gon et al., 2013; Amato et al., 2014). Are the authors planning to include these emission processes in the CARS system as part of future developments? If so, it may be good to mention it in the conclusions section (or at least mention the current limitation of the system regarding the estimation of PM emissions).*

*Hugo A.C. Denier van der Gon, Miriam E. Gerlofs-Nijland, Robert Gehrig, Mats Gustafsson, Nicole Janssen, Roy M. Harrison, Jan Hulskotte, Christer Johansson, Magdalena Jozwicka, Menno Keuken, Klaas Krijgsheld, Leonidas Ntziachristos, Michael Riediker & Flemming R. Cassee (2013) The Policy Relevance of Wear Emissions from Road Transport, Now and in the Future—An International Workshop Report and Consensus Statement, Journal of the Air & Waste Management Association, 63:2, 136-149, DOI: 10.1080/10962247.2012.741055*
*Fulvio Amato, Flemming R. Cassee, Hugo A.C. Denier van der Gon, Robert Gehrig, Mats Gustafsson, Wolfgang Hafner, Roy M. Harrison, Magdalena Jozwicka, Frank J. Kelly, TeresaMoreno, Andre S.H. Prevot, Martijn Schaap, Jordi Sunyer, Xavier Querol, Urban air quality:The challenge of traffic non-exhaust emissions, Journal of Hazardous Materials, 275, 31-36, https://doi.org/10.1016/j.jhazmat.2014.04.053, 2014.*

Thanks for those comments. Korea National Institute of Environmental Research (NIER) currently does not have a set of emission factors for PM from tire and break wears yet. However, the CARS model is designed to process any pollutants form any process, such as tire and break wears as long as the emission factors are available in the emission factors input file. Once those emission factors are available, the CARS model can estimate the tire and brake wear emissions. The Line 650 to 654 has been modified and add those two references, now it reads:

"The current South Korea NIER currently does not have the PM emission factors from tire and brake wear, which are the highest contributors of $PM_{2.5}$ emissions from onroad vehicles (Hugo A.C. et al., 2013; Fulvio Amato et al., 2014). Once the emission factors of tire and brake wear are prepared, those emissions can be computed by CARS."

*The CARS system considers the influence of temperature on different emission processes (e.g., cold-start, NOx diesel hot exhaust). How is the information of temperature provided to the CARS system by the user? Can the user provide gridded information? Or only a single set of temperature values for the whole domain of study? Please specify in the text.*

There are three parameters ("temp_max", "temp_mean" and "temp_min") in CARS model that allows users to define the temperature settings in the CARS runs. Those temperature parameters are for all model domain for the simulation period. Current version of CARS does not support to process gridded meteorology data for onroad mobile emission calculation yet. However, the user can simply adjust the temperatures by day, month, or season. to generate the appropriate temporally resolved emissions. We clarified this in our text in model configuration part in Line ==453 to 459==. Now it reads:

"The influence of temperature on emission processes are considered in the CARS model. There are three temperature parameters in current CARS model such as "temp_max" for maximum temperature, "temp_mean" for mean temperature, and "temp_min" for minimum temperature. These temperature parameters will be applied to over the entire modeling domain during the simulation period. Current CARS model version does not support to process gridded meteorology data from the 3$^{rd}$ party meteorology models like Meteorology-Chemistry Interface Processor (MCIP) from U.S. EPA., and Weather Research Forecasting (WRF) model from National Center for Atmospheric Research (NCAR) yet. However, CARS can easily adopt various temporally resolved temperature values by adjusting the CARS simulation period (i.e., day, week, month, season, or annual)."

*The CARS system is capable of computing CTM-ready emission inputs. Could you provide a list of the CTMs that are currently compatible with the CARS output files (e.g., CMAQ, WRF-CHEM,...)? (for each CTM, emission input files need to be provided ina specific format, e.g., attributes and name of the variables of the NetCDF file, spatial projection, units)*

Thanks for these comments. The current version of CARS only support CAMQ-ready gridded hourly emissions. The line ==512 to 514== has bee modified and now it reads:

"It should be noted that current CARS model can only generate the Community Multiscale Air Quality (CAMQ)-ready gridded hourly emissions in format of IOAPI (Input/Output Applications Programming Interface) based on NetCDF format."

*While Figure 1 of the manuscript gives a clear overview of the CARS methodology and workflow, I think it would be good to also include a summary table with a list of the names of the input files that are needed to run the system, classified by category (i.e., activity data, emission factors, ...).*

Thanks for these comments, we add Appendix H to show the summary of input files, and a sentence has been added in Line ==159 to 160==. Now it reads:

"The summary of input files by categories are presented in Appendix H."

*I recommend to update the reference Rey DR (2018) to Rodriguez-Rey et al. (2021): Rodriguez-Rey, D., Guevara, M., Linares, MP., Casanovas, J., Salmerón, J., Soret, A.,Jorba, O., Tena, C., Pérez García-Pando, C.: A coupled macroscopic traffic and pollutant emission modelling system for Barcelona, Transportation Research Part D, 92, https://doi.org/10.1016/j.trd.2021.102725, 2021.*

Thanks for this comment, the reference has been updated to the latest version for Rodriguez-Rey et al (2021)

**Referee #2 Comments**

*General Comments*
*Development of a new open-source vehicle emissions model has merit. However, I am concerned about the rationale in using the average speed distribution from the State of Georgia in use in this study, also with the language suggesting that it is an improvement in accuracy. I am concerned there is a disconnect in the use of average speed distribution in MOVES and in the CARS model, and don't understand why an average speed distribution could be calculated from the link-level data from South Korea's GSI road shape files. I added substantial questions and comments regarding this issue in the specific comments below. (See Page 8. Line 271-281).*
*Because I felt this issue must be addressed before going forward with the paper, I stopped my review at this point. And only conducted a cursory review of the remaining aspects of the paper, including results and conclusion.*
*I have also added many specific comments to remove generalizations that may be not accurate, provide citations behind some of the statements, and to clarify the calculations which I encountered in the abstract, introduction and methods section.*
*I am willing to re-review the paper if my concerns can be addressed regarding the average speed distributions in the model.*

We thank our reviewers for their constructive comments on our model development. Those critical comments improve our manuscript stronger. As the reviewer points out, there are considerable uncertainties around the average speed distribution. Here, we followed the referee's comments and have modified the manuscript accordingly.

***Specific comments*** *(suggested text in italics)*
*p.1 Line 24 "it can optionally utilize road link-specific average speed distribution (ASD)" - is it an average speed distribution of the road type, or the individual link? The wording is not clear. Should it be referred to road-specific average speed distribution? Like on p.1 line 32?*

Thanks for the comment. We would like to clarify that the ASD here is average speed distribution of each road type, and the data is from the road shape file of South Korea. The Line ==24 to 25== has been modified and now it reads:

"It can optionally utilize average speed distribution (ASD) of all road types to reflect more realistic vehicle speed variations."

*p.2 Line 39-41 "It indicates that the CNG bus is better for the rural area while the diesel bus is better applicable for the urban area for a better ozone control strategy because the rural area is usually NOx limited for ozone formation and urban area is VOC limited region" Is this backed up with air quality modeling, or assumed based on the reasons given here? In practice, couldn't it be much more complex? If it is not built on analysis, then I think the statement should be re-written as a potential ozone control strategy—which may need to be backed up with more analysis. E.g. what would be the impact on suburban areas of more NOx or VOC emissions?*

Thanks for this comment. We understand the complicated ozone formation between limited NOx and VOC relations. The potential ozone impact part is not appropriate in this paper. The Line ==38  to 39== has been modified, now it reads:

"In VOC emission part, CNG buses are the largest contributor 39 with 19.5% of total VOC emissions."

*p.2. line 47. The line about indoor vs. total air pollution makes it seem that ambient air pollution is a relatively minor contributor to public health. I would add more citations to clarify that ambient pollution impacts indoor air quality, or remove the indoor air quality references—as potentially misleading. For example: Cohen et al. 2017 estimate 4.2 annual early deaths to ambient PM. Cohen, A. J., et al. (2017). Estimates and 25-year trends of the global burden of disease attributable to ambient air pollution: an analysis of data from the Global Burden of Diseases Study 2015. The Lancet, 389 (10082), 1907-1918. DOI: https://doi.org/10.1016/S0140-6736(17)30505-6. Burnett, et a. 2018 estimate the health burden is closer to 9 million deaths from ambient PM concentrations Burnett, R., et al. (2018). Global estimates of mortality associated with long-term exposure to outdoor fine particulate matter. Proceedings of the National Academy of Sciences, 115 (38), 9592-9597. DOI: 10.1073/pnas.1803222115.*

Thank you for this comment. The Line ==45 to== 47 has been modified and add the new references, now it reads:

"Globally, ambient pollution causes more than 4.2 million premature deaths every year (Cohen et al., 2017), and Burnett et al. estimate the health burden is closer to 9 million deaths from ambient PM concentrations (Burneet et al, 2018)."

*p. 2. Line 57-59. Is this statement backed up with a citation? If not, I would not say this could be an over-generalized statement is generally always accurate. For example, in areas with persistent cold pooling inversions—modeling the meteorology and chemistry may just or more critical then accurately modeling the emissions—correct?*

Thank you for this comment. This overall-generalized statement part has been removed.

*p.2 60-61. Another over-generalized statement that deserves more context and a citation. For which pollutant? Do you mean for NOx? I don't think this is true for VOCs, and was a little surprised that it was mentioned as true for PM2.5 (because that is not the case with the US NEI for select urban counties).*

Thanks for this comment. The Line ==53 to 54== has been modified, now it reads:

"The transportation emission sector is one of the major anthropogenic emissions in urban areas."

*p. 2 lines 67-69. Are these studies based on air quality modeling or observations? Either way, I would suggest these results are presented in the context of other studies that show that primary PM2.5 is a minor contributor to ambient PM2.5.*
*Nault et al. 2021 suggests that PM2.5 in Seoul (and other major urban areas across the world) have only a small contribution of PM2.5 from primary PM2.5. Most is secondary formed PM either as secondary organic aerosol or secondary inorganic PM2.5 (ammonium-sulfate or ammonium-nitrate).*
*Nault, B. A., et al. (2021). Secondary organic aerosols from anthropogenic volatile organic compounds contribute substantially to air pollution mortality. Atmos. Chem. Phys., 21 (14), 11201-11224. DOI: 10.5194/acp-21-11201-2021.*
*See also*
*Jimenez, J. L., et al. (2009). Evolution of Organic Aerosols in the Atmosphere. Science, 326 (5959), 1525-1529. DOI: 10.1126/science.1180353.*

Thanks for this comment and all the provided references, we have modified our sentences. The Line ==57 to 61== has been modified and add those reference, now it reads:

"In the Seoul Metropolitan Area (SMA) in South Korea, transportation automobile sources contribute the most to the total $NO_X$ and primary $PM_{2.5}$ emissions across all emission sources. (Choi et al., 2014; Kim et al., 2017a; Kim et al., 2017b; Kim et al., 2017c)."

*p. 3. Line 73. " highly resolved spatiotemporal automobile emissions"*
*- bottom-up emissions inventories by process can give high resolution, but are not necessarily higher quality than top-down methods.*

*Aren't the vehicle operation processes both physical and chemical? I would recommend stating that you can get more spatiotemporally resolved emissions inventories when pairing process-specific emissions models with resolved vehicle activity data.*

Thanks for this comment. The Line 62 to 64  has been modified, now it reads:

"The use of process-based automobile emission models is highly recommended to meet the needs in CTM model because it can estimate the highly resolved spatiotemporal automobile emissions."

*p. 3. Line 97-98. They are developed differently to meet specific user needs? their own needs based on the types of traffic activity and emission factors, emission calculation methodologies, and other optional/available.*
*- I hope the emissions models are not developed to meet the models own needs….but on the model users needs*
*- Should you mention that each model is developed with different levels of specificity, underlying data sets, and modeling assumptions?*

Thanks for this comment. The Line 86 to 89 has been modified, now it reads:

"They are developed differently to meet specific user needs based on the types of traffic activity and emission factors, emission calculation methodologies, and other optional/available traffic-related inputs such as average speed distribution and geographical resolution. Each model is developed with different levels of specificity, underlying data set and modeling assumptions."

*p.3. lines 100-101. This statement is not clear, and not sure it is needed here.*

Thanks for this comment. This sentence has been removed.

P. 108-109. I disagree about the general statement on the lack of transparency for emission factors. Technical reports that document the emission factors and algorithms for estimating emissions are available here:
https://www.epa.gov/moves/moves-onroad-technical-reports
Thanks for this comment. The Line 95 to 97 has been modified, now it reads:

"Disadvantage of this model is it difficult to update and apply to countries outside of the U.S. because MOVES model is high degree of specificity."

*Page 6. Line 180-181. What do you mean by traffic density? Are you referring to total VKT as the subtitle suggests? Isn't VKT a measure of traffic flow rather than density?*

Thanks for this comment. The Line 168 to 170 has been modified, now it reads:

"The individual vehicle VKT data is used to reflect the human activity. This study imported the national registered vehicle-specific daily total VKT from South Korea's Vehicle Inspection Management System (VIMS), which belongs to the Korea Transportation Safety Authority (KTSA)."

*Page 6. Line 192-193. How is the VIN used in the calculation? Is that to calculate the vehicle age? Could you clarify?*

Thanks for this comment. The Line 179 to 180 has been modified, now it reads:

"The VIN ($vin$) information is used to calculate vehicle-specific daily average VKT ($VKT_{vin}$, km d$^{-1}$)."

*p. 6 line 194-195. VKT with the manufactured year (VKTv,age) is calculated based on the cumulative mileage (Mf)*  *between the last inspection date (Df) and registration date (D0).*
*- Clarification on the data in the calculation. Is the registration date always at or near age zero?*
*Does the registration data only capture vehicles when they change ownership?*

*Or does that happen more regularly?*
*I think it is important to clarify if the VKT calculated in equation 1, is reflective of most recent years of use, or an average VKT over the lifetime of the vehicle.*
*- Also, since equation 1 is applied to individual vehicles, shouldn't the subscript reflect individual vehicle in the equations, rather than just vehicle type, and age?*

Thank you for this comment. Here are our responses:

1. Yes, the information of registration date is close to age zero with high probability, and the date of initial registration is the D0. Therefore, one vehicle has only one initial registration date and it won't be affected by the change of vehicle ownership.
2. The subscript $v$ of VKT, Mileage, and Date were not clear. So, we have updated the subscript. VIN for individual vehicle is now "$vin$". Thus, the VKT in Eq.1 is the daily total VKT over the lifetime of the individual vehicle.

We have modified this paragraph (line 179 to 186) and Eq.1 and now it reads:

"The VIN ($vin$) are applied to individual vehicles to calculate their daily average VKT ($VKT_{vin}$, km d$^{-1}$). In Eq. (1), the individual vehicle daily average VKT ($VKT_{vin}$) is calculated based on the cumulative mileage ($M_{f;vin}$) between the last inspection date ($D_{f;vin}$) and registration date ($D_{0;vin}$). Each vehicle is categorized with Korea's NIER defines the vehicle types (Ryu et al., 2003; Ryu et al., 2004; Ryu et al., 2005; Lee et al., 2011a) that based on a combination of vehicle types (e.g., sedan, truck, bus, etc), engine sizes (e.g., compact, full size, midsize, etc) and fuel types (e.g., gasoline, diesel, LPG, etc). Full details of vehicle types and daily total VKT are shown in Appendix A and B."

$$VKT_{vin} = \frac{M_{f;vin}}{D_{f;vin} - D_{0;vin}} \qquad (1)$$

*p. 6 lines 205-207. 'nonroad automobile' is potentially confusing. Vehicles in MOVES are classified as onroad or nonroad. When passenger vehicles are operating in driveways, parking lots, I would classify this as off-network automobile emissions.*
*Can you clarify what are the off-network automobile emissions that are missing? Are those starts? Evaporative? Idling emissions? Or are they spatially allocated to the roadways.*

Thanks for this comment. The Line 189 to 193 has been modified, now it reads:

"Automobile emission sources include motorized engine sources on the paved road network including off-network (e.g., drive way and parking lots). The CARS model doesn't simulate emissions from nonroad emission sources, such as aviation, railways, construction, agricultures, lawn mower, and boats yet. The CARS model simulates the onroad automobile emissions from network roads using their local traffic-related datasets."

*Line 225-227. Should you also mention that emissions are produced from incomplete combustion products that are not controlled from the emissions aftertreatment equipment, such as a three-way catalytic converter for gasoline vehicles?*

Thanks for this comment. The Line 212 to 215 has been modified, now it reads:

"The ICE combustion cycle generally causes incomplete combustion processes which emit hydrocarbons, carbon monoxide (CO), and particulate matter (PM) which are not completely controlled from the aftertreatment equipment, such as three-way catalytic converter and released into atmosphere. "

*Line 228- I would not say NOx is similarly produced as SOx, because the source of S is the fuel, not the atmosphere like nitrogen.*

Thanks for this comment. The Line 216 to 217 have been modified, now it reads:

"Nitrogen oxides (NO$_x$) are produced during the combustion process due to the abundant nitrogen (N$_2$) and oxygen (O$_2$) in the atmosphere. "

*Page 7. Line 280-235. Does the age in the emission factor, reflect the impact of model year on the new vehicle emission rate, and age in the DF factor reflect aging effects from the new vehicle emission rate? If so, could that be clarified—otherwise, it is not clear what the purpose of DF is. However, if everything is calculated in terms of age— does that mean an emission rate for a 5 year-old vehicle is the same, for all calendar years? Or does model only work for one calendar year?*
*Page 8. 247-248. In this equation, is vehicle age used to reflect the model year or technology effect of a new vehicle, or a deterioration effect? If everything is in terms*

Thank you for this comment. Please see our responses:

1. In previous version, the parameter "age" in this section may confuse the readers, so *age* is now replaced by manufacture year (*myr*). This *myr* variable is listed in both emission factor table and deterioration table. The vehicle age is the internal variable calculated based on the targeted simulation year and the individual vehicle manufacture year (*myr*). This calculated vehicle age is used to determine the age-specific DF factors (up to 16 years old) in the DF process.
2. The vehicle emission factors are decided by the parameters including "manufacture year (*myr*), pollutants (*p*), vehicle type (*v*), and vehicle speed (*s*)."
3. The DF are used to reflect the emission increase caused by vehicle aging in the model. The DF are decided by parameters including "manufacture year (myr), pollutants (*p*), vehicle type (*v*), and vehicle age (from 1 to 16)". The age 16 DF will be applied to any vehicles older than 16-year-old.

For example, If CARS models a 2001 sedan gasoline vehicle with compact engine for year 2015, CARS will first seek e emission factors for the vehicle by manufacture year (*myr: 2001*), the vehicle type (*v*), the speed data (*s*), and pollutant (*p*). The DF will be also found by the manufacture year (*myr*), vehicle type (*v*), pollutants (*p*) and the vehicle age (14-year-old). Then, the daily total emission rate of this vehicle will be computed with *VKT$_{vin}$* multiplied by the DF and emission factors.

The Line 224 to 240 has been modified, now it reads:

"The deterioration factor (*DF*) in Eq. (3) is an optional function in the CARS model. This deterioration process is caused by vehicle aging and can lead to the increase of vehicle emissions. The vehicle deterioration factor is varied by vehicle type, pollutant, vehicle manufacture year and vehicle age. The CARS model applies the vehicle manufacture year (*myr*) and model simulation year to calculate the age of vehicle. According to the guidance of deterioration factors calculation from NIER, there is no deterioration in a new vehicle in their first five years. After five years, the deterioration factors can increase the range by 10% depending on the type of vehicle and pollutants. Deterioration processes can cause a 50% or 100% increase of emissions in fifteen-year-old vehicles. Currently, the *DF* is an empirical coefficient that varies by vehicle age (Lee et al., 2011a).
The hot exhaust emission factor, *EF$_{hot;p,v,s}$* (g/km) is a function of vehicle speed (*s*) with other empirical coefficients: *a, b, c, d, f, k*. The emission factor formula and those coefficients were developed by NIER CAPSS (Lee et al., 2011a). These coefficients are varied by pollutants (*p*), vehicle type (*v*), vehicle manufacture year (*myr*), and vehicle speed (*s*). The vehicle speed affects the combustion efficiency of an ICE and impacts the emission rates and its composition from the tailpipe.

$$EF_{hot; p,v,myr,s} = k(a \times s^b + c \times s^d + f) \qquad (4)"$$

*Page 8. 255 "Figure 3a shows a significant decrease of NOx emissions while speed increases between 0 and 70 kmh.*

Thanks for this comment. The Line 245 to 246 has been modified, now it reads:

*"Figure 3a shows a significant decrease of NO$_x$ emissions while speed increases between 0 and 70 km."*

Page 8. Line 260. I would remove word 'constant' speed. That implies that the vehicle emissions model can differentiate between emission factors between constant speeds, and transient sec/sec speeds, which I do not think is the case. My understanding is that you are differentiating between a single average speed, to an average speed distribution. However, the associated emission rates with the average speeds is not changing.

*Page 8. Line 262. Remove 'incomplete ICE combustion' higher NOx emissions are not necessary directly related to an issue with incomplete combustion—NOx emissions can be lower when ICE conditions are fuel rich. I would just recommend that a single speed, may not represent the average emission rates, as an average speed distribution with time spend at multiple speeds.*

Thanks for this comment. Yes, those are true, the "constant" and "incomplete ICE combustion" have been removed.

The Line 248 to 249 has been modified, now it reads:

"When a single speed is assigned to compute hot exhaust emissions, it won't reflect the emissions under low-speed circumstances."

*Page 8. Line 271-281.*
*I believe there may be an issue with using ASD from MOVES for Korea in CARS.*

*1. Average speed distributions from MOVES and provided from the State of Georgia should be defined. MOVES average speed distributions are intended to calculate the distribution of average speeds across multiple links within the same road type. As well, as to capture the distribution in average speeds in links across time (such as days with traffic incidents or normal travel days). In practice, the data is aggregated from telematics data that calculate average speeds from varying resolutions (1 hz to every 180 seconds). As documented in the MOVES population and activity report.*
*USEPA (2020). Population and Activity of Onroad Vehicles in MOVES3. EPA-420-R-20-023. Office of Transportation and Air Quality. US Environmental Protection Agency. Ann Arbor, MI. November 2020. https://www.epa.gov/moves/moves-technical-reports.*
*I am not familiar with how the State of Georgia average speed distributions were calculated, but that should be explained if those are used in the study.*

*2. Why not develop average speed distributions from South Korea's GSI road shape files? The text seems to suggest that it is a problem that there is only one average speed associated with each road link. But this is not a problem. There should be many links of the same road type within a region. Using that data, you can calculate an average speed distribution for that road type in the region. Is it because you want to capture variation in link levels average speeds across different days or hours of the day? If so please clarify.*

Thank you for these comments. Please review our combined responses:

The CARS model is designed to process the ASD by road types. The reason that we considered SK data and George data in this study is because the link-level speed data from South Korea in this study is limited to a single average speed value per link, while US link-level average values are based on the telematics. Therefore, there is no variation speed pattern for each road link. Figure 4a is based on the original link-specific average speed value from the SK shapefile. These ASD by road type does not represent a proper cycle of average speed distribution from a road link, because they are simply a collection of average speed values from road links. It lacks on representing the low speed bins which impacts the most of many pollutants due to the incomplete engine combustion. Compared to Figure 4b which is the ASD based on the telematics from George, the low speed bin representations from SK was incorrect.

We are currently working with other researchers from Korea to develop the Korea ASD dataset based on their own local measurements. Once we develop the Korea-specific ASD, we can regenerate the updated onraod mobile emissions. In this section, we want to point out the CARS's functionality and capability, as well as the impacts of ASD to local emissions inventory from onroad mobile sources than the ASD input data issue itself.

We believe that the performance of CARS will be improved once we incorporate their own ASD into the CARS simulation.

The Line 259 to 274 has been modified to clarify this part, now it reads:

"We first developed the ASD (Fig. 4a) for eight different road types (No. 101-108) in South Korea based on the latest road link-specific average speed and the length of link from the SK GIS road network shapefiles (NIER, 2018). Because the original link-level speed data is averaged, we used the link length as a weighting factor to show the variation of speed pattern for each link. However, the ASD based on the SK GIS road shapefiles wasn't able to capture the low-speed range (<16 km h$^{-1}$) that occurs while it operates (Fig. 4a). It caused the significant underestimation of NOx and VOC emissions compared to the CAPSS (Appendix G).

To address this SK ASD issue, we incorporated the ASD (Figure 4b) from the state of Georgia developed by U.S. EPA to improve the representation of the low-speed ranges (speed bin #1 and #2 for road type 1 to 7). We increased the total fractions of low-speed bins (the 2:1 ratio of fractions of bin #1 and #2) by 2% for interstate expressways, 3% for urban expressways, 7% for all highways, and 15% for all local roads. The increases in low-speed bins lowered the distributions of other higher speed bins homogeneously due to the renormalization of fractions by road type. Figure 4c shows the renormalized hybrid-ASDs of all road types based on SK ASD and Georgia ASD. We understand, the hybrid-ASD approach is not ideal for SK onroad emission inventory development. However, it clearly demonstrates the CARS's capability and sensitivity to the vehicle speed representation and the impacts of ASD to the local onroad mobile inventories."

*3. It is not mentioned if the average speed distributions vary by time of day—please clarify, they do in MOVES, which captures the effect of the diurnal traffic pattern on vehicle speeds.*

The current version of CARS does not support time-dependent ASD profiles yet. We will update the manuscript to clarify this limitation. See the updated manuscript below in line 256 to 258:

"Although ASD patterns vary by region and time, current CARS model version does not support ASD application by region and time of day due to the lack of region and time dependent ASD availablity in South Korea "

*4. The emission rates from the CARS model—are they intended to be associated with a cycle average speed that represents the average speed with a road link (which contains variation of speed within the link)? Or are the emission rates intended to be associated with sec/sec speed data? If the emission rates are intended to be cycle or link average emission rates, then it makes sense to use average speed distribution calculated from the average speeds from many links, rather than using an average speed distribution calculated from sec/sec data.*

Thanks for this comment. Yes, the CARS model intended to be associated with a cycle average speed represent the average speed with a road link. But the link level speed data in SK are average speed only, there is no variance speed by link and by road type. We explained the details in the previous response. We have consulted with Korean's collaborators, but we were not able to locate any appropriate dataset to develop the SK ASD by road types for CARS modeling runs.

*5. Is using the average speed distribution approach even needed here? If you have the average speed for each individual road link? Why not use that approach? Add an explanation.*

We computed the emissions based on the ASD from Figure 4a, and compared to the Korea NIER CAPSS and found the results are significantly underestimated NOx and VOCs due to the poor representation of vehicle during the computation. See the Appendix G of emissions comparison between CAPSS, original ASD and the adjust ASD approach.

*6. It is not clear why using ASD from the state of Georgia are more realistic than the using the current data. Also, the explanation is not clear on the development of the inputs. E.g. where did the 2:1 ratio of bins 1 and 2 come from. And where did the additional 2%, 3%, 7%, and 15% come from. Was it added such that the average speeds on each roadway is the same before and after the fix?*

Thank you for this comment. We don't think Georgia data are more realistic than Korea data, but Korea data missed the link-level speed variation information and cause the missing of low-speed profile. Therefore, we consider Georgia state's data and make some assumption for the low-speed bins. We understand this hybrid-ASD based on SK and Georgia ASDs is not perfect, but we think this is the best approach at this time.

*7. To me, it seems like this should be better classified as a 'sensitivity' study on the potential impact of using average speed distributions. Since, the data for the average speed distributions used in the study, don't seem to be clearly better than what is there, and seem to be based on quite a few assumptions.*

Thank you for this comment. We agree with your suggestion. This is a model development study, not the sensitivity test study. We don't have to adjust for the original input data or did the sensitive test. On the other hand, current CARS model is a research model, and as a model developer, we hope the CARS model result can be compared with current South Korea official model system, Korea Clean Air Policy Support System (CAPSS).

***Technical corrections:** (suggested text in italics)*
*p. 1. Line 18. "utilize the local vehicle activity data"*

Thank you for this correction, we have modified it.

*p.1 Line 20 "to generate a temporally and spatially resolved "*

Thank you for this correction, we have modified it.

*p. 2 line 28, "due to  it having the longest daily VKT and relatively high NOx g/km emission rate."*

Thank you for this correction, we have modified it.

*p. 4. Line 133 "changing specific variables that may be . embedded in the code?"*

Thank you for this correction, we have modified it.

*p.5 line 151 "road link-level"*

Thank you for this correction, we have modified it.

*p.5 line 174. "South Korean traffic databases from "*

Thank you for this correction, we have modified it.

*line 224 "hot exhaust emissions"*

Thank you for this correction, we have modified it.

*p. 46 Appendix E and F. Speed units should be km/hour. Also, what is the range of the speed bins? That is not clear from the Tables.*

Thank you for this correction, we have added the speed range for each speed bins.

[revised manuscript text omitted]

---

## Referee Report (RR1)

Major comments:

Line 645-646. In your comments you mention that it was not appropriate to talk about ozone control. However I still see this statement here. Please remove

Lines 259-274.

I disagree with the statement in the author's response which states:

"These ASD by road type does not represent a proper cycle of average speed distribution from a road link, because they are simply a collection of average speed values from road links."

The average speed distribution in MOVES is intended to be a collection of average speed values from different roadway links.

You could mention that the average speeds from SK, don't include the lower average speeds that are anticipated to occur across time due to congestion. However, I can see that the authors wanted to evaluate the sensitivity to having average speeds with more lower speed driving, and You

I don't think it is defensible for you to say that the adjustments from the State of George is better, or that the current values are underestimated. I think you can say that you obtain different values, by adjusting the SK ASD values based on the State of Georgia ASD. And you state your assumptions about why you made the adjustments.

I would be ok with the adjustments to the SK ASD, if you clarified that . I have suggested edits below.

> However, the ASD based on the SK GIS road shapefiles wasn't able to did not capture the low-speed range (<16 km h-1) driving that occurs while it operates (Fig. 4a). It This causes d the a significantly under lower estimation of NOx and VOC emissions compared to the CAPSS (Appendix G). We believe the SK average speed distribution is missing low speed driving that can occur on links on different days due to congestion. To address this absence of low-speed driving in the SK ASD issue, we incorporated data from the ASD (Figure 4b) from the state of Georgia developed by U.S. EPA to improve the representation adjust the low-speed ranges (speed bin #1 and #2 for road type 1 to 7).

In response to comment 4. You mention "But the link level speed data in SK are average speed only, there is no variance speed by link and by road type."

I don't understand this statement. This makes me think that you have a different definition of average speed distribution than MOVES. Looking at Figure 4a, you clearly have calculated an average speed distributions for each average speed from each road type and link from the SK data. To calculate the Figure 41, you clearly have average speed by link that varies within each roadtype.

These average speed distributions from each link are appropriate for use in MOVES. But then you mention that it is not sufficient. Are you thinking that you need speed distributions based on sec/sec data? Rather than based on average link-level speeds?

I am ok with your changes. But want to make sure you're giving the correct rationale for making the changes. To me it seems that you want to calculate average speed distributions across links, road types

AND time. You have links and road types. It seems that the missing dimension you would like to have in your current average speed distribution is time. You want to calculate an average speed distribution based on link-level average speed that vary across time due to days with congestions.

Lines 695-701

I'm good with the description here between average speed distributions and single speeds—and your assertion that the average speed distribution is better

Minor changes: (clarify text)

Line 95-96 "MOVES has a high degree of specificity"

Line 179-186:

Each vehicle is categorized with Korea's NIER which defines the vehicle types (Ryu et al., 2003; Ryu et al., 2004; Ryu et al., 2005; Lee et al., 2011a)  based on a combination of….

Line 189-190

Recommend clarifying that you have both on and off-network sources. The current text implies that off-network is part of on-network??

Automobile emission sources include motorized engine sources  on the paved road network  and off the road network (e.g., drive way and parking lots).

Line 216 to 217

Recommend changing this to the engine—fuel rich engine conditions don't produce high amount of NOx. But NOx is produce in lean-burn conditions

"Nitrogen oxides (NOx) are produced due to the  abundance of nitrogen (N2) and oxygen (O2)  during the combustion process . "

---

## Author Response (AR2)

**Responses to the Reviewer's Comments**

We would like to thank the reviewers for their comments and believe that it has improved the manuscript. The reviewers' comments are in grey italics and our response is given in black.

Line 3 to 14 has been updated to reflect the recent changes in the author's affiliation changes.

**Referee Comments:**

*Line 645-646: In your comments, you mention that it was not appropriate to talk about ozone control. However I still see this statement here. Please remove*

Thanks for this comment. Line 645 to 646 has been removed.

*Lines 259-274: I disagree with the statement in the author's response which states:*

*"These ASD by road type does not represent a proper cycle of average speed distribution from a road link, because they are simply a collection of average speed values from road links."*

*The average speed distribution in MOVES is intended to be a collection of average speed values from different roadway links.*

*You could mention that the average speeds from SK, don't include the lower average speeds that are anticipated to occur across time due to congestion. However, I can see that the authors wanted to evaluate the sensitivity to having average speeds with more lower speed driving, and*

*I don't think it is defensible for you to say that the adjustments from the State of George is better, or that the current values are underestimated. I think you can say that you obtain different values, by adjusting the SK ASD values based on the State of Georgia ASD. And you state your assumptions about why you made the adjustments.*

*I would be ok with the adjustments to the SK ASD, if you clarified that . I have suggested edits below.*

> *However, the ASD based on the SK GIS road shapefiles  did not capture  low- speed range (<16 km h-1) driving  (Fig. 4a). It This causes  significantly  lower estimation of NOx and VOC emissions compared to the CAPSS (Appendix G). We believe the SK average speed distribution is missing low speed driving that can occur on links on different days due to congestion. To address this absence of low-speed driving in the SK ASD , we incorporated data from the ASD (Figure 4b) from the state of Georgia developed by U.S. EPA to  the low-speed ranges (speed bin #1 and #2 for road type 1 to 7).*

*In response to comment 4. You mention "But the link level speed data in SK are average speed only, there is no variance speed by link and by road type."*

*I don't understand this statement. This makes me think that you have a different definition of average speed distribution than MOVES. Looking at Figure 4a, you clearly have calculated an*

*average speed distributions for each average speed from each road type and link from the SK data. To calculate the Figure 41, you clearly have average speed by link that varies within each roadtype.*

*These average speed distributions from each link are appropriate for use in MOVES. But then you mention that it is not sufficient. Are you thinking that you need speed distributions based on sec/sec data? Rather than based on average link-level speeds?*

*I am ok with your changes. But want to make sure you're giving the correct rationale for making the changes. To me it seems that you want to calculate average speed distributions across links, road types AND time. You have links and road types. It seems that the missing dimension you would like to have in your current average speed distribution is time. You want to calculate an average speed distribution based on link-level average speed that vary across time due to days with congestions.*

We really appreciate your accurate comments and suggestions. You are correct. We did not properly state in our responses on ASD development based on the SK road shapefiles. While the average speed values from reach road type and link from SK road data can be used for ASD development for use in MOVES, it did not represent actual low-speed driving patterns occurring during traffic congestions.

Following your suggestion. Line 264-272 has been modified. Now it reads:

"We first developed the ASD (Fig. 4a) for eight different road types (No. 101-108) in South Korea based on the latest road link-specific average speed and the length of link from the SK GIS road network shapefiles (NIER, 2018). However, the ASD based on the SK GIS road shapefiles did not capture low-speed range (<16 km h$^{-1}$) driving (Fig. 4a). This causes a significantly lower estimation of NOx and VOC emissions compared to the CAPSS (Appendix G). We believe the SK average speed distribution is missing low-speed driving that can occur on links on different days due to traffic congestion. To address this absence of low-speed driving in the SK ASD, we incorporated data from the ASD (Figure 4b) from the state of Georgia developed by U.S. EPA to the low-speed ranges (speed bin #1 and #2 for road type 1 to 7)."

*Lines 695-701: I'm good with the description here between average speed distributions and single speeds—and your assertion that the average speed distribution is better*

Thanks for your response to our updates.

*Line 95-96 "MOVES has a high degree of specificity"*

Thanks for this comment. Line 101-102 has been modified. Now it reads:

"Disadvantage of this model is it difficult to update and apply to countries outside of the U.S. because MOVES model has a high degree of specificity."

*Line 179-186: Each vehicle is categorized with Korea's NIER which defines the vehicle types (Ryu et al., 2003; Ryu et al., 2004; Ryu et al., 2005; Lee et al., 2011a)  based on a combination of....*

Thanks for this comment. Line 187-191 has been modified. Now it reads:

"Each vehicle is categorized with Korea's NIER which defines the vehicle types (Ryu et al., 2003; Ryu et al., 2004; Ryu et al., 2005; Lee et al., 2011a) based on a combination of…."

*Line 189-190: Recommend clarifying that you have both on and off-network sources. The current text implies that off- network is part of on-network??*

*Automobile emission sources include motorized engine sources  on the paved road network  and off the road network (e.g., drive way and parking lots).*

Thanks for this comment. Line 194-195 has been modified. Now it reads:

"Automobile emission sources include motorized engine sources on the paved road network and off the road network (e.g., drive way and parking lots)."

*Line 216 to 217: Recommend changing this to the engine—fuel rich engine conditions don't produce high amount of NOx. But NOx is produce in lean-burn conditions*

*"Nitrogen oxides (NOx) are produced due to the  abundance of nitrogen (N2) and oxygen (O2)  during the combustion process . "*

Thanks for this comment. Line 221 to 222 has been modified. Now it reads:

"Nitrogen oxides ($NO_x$) are produced due to the abundance of nitrogen ($N_2$) and oxygen ($O_2$) during the combustion process.

---

## Author Response (AR4)

**Responses to the Reviewer's Comments**

We would like to thank the reviewers for their comments and believe that it has improved the manuscript. The reviewers' comments are in grey italics and our response is given in black.

**Referee Comments:**

*Dear Authors,*

*Thank you for improving the language of the manuscript. I still need to ask you to modify the code availability section. You are still referring to github website which is not a permalink. Please remove this and simply have text reading something like "The CARS Version 1 used in the paper is provided by Baek et al. (2021) ". Then have the reference in the reference list. You also have website addresses in User manual and installation package sections whereas at least the first is also included into the Zenodo set. Any non permalink websites should be avoided as is clearly reading in GMD Editorial. Also installation package shoudld be there.*

*Reference is in format*

*B.H. Baek, Rizzieri Pedruzzi, & Chi-Tsan Wang. (2021). bokhaeng/CARS: CARS (Comprehensive Automobile Emissions Research Simulator) version 1.0 Public Release (CARSv1.0). Zenodo. https://doi.org/10.5281/zenodo.5033314*

Thanks for this comment. Code availability section and references has been Changed.

Lines 282-283: The details of the road restriction control table format can be found on the CARS's user's guide from the CARS

version 1 used in this paper (Baek et al., 2021).

Lines 728-748:

**Code Availability:**
The source code of the CARS model public release version 1.0 can be downloaded from the Github release website:

https://doi.org/10.5281/zenodo.5033314

**Digital Object Identifier (DOI) for the CARS version 1.0:**

https://zenodo.org/record/5033314#.YNzDrC1h001

https://doi.org/10.5281/zenodo.5033314

**Installation Package for CARS version 1.0:**

The CARS version 1.0 installation package comes with the complete inputs and outputs datasets for users to confirm their proper installation on their computers and can be downloaded from the Github release website:CARS version 1 used in this paper (Baek et al., 2021):

https://github.com/bokhaeng/CARS/releases/download/CARSv1.0/CARS_v1.0_public_release_package_25June2021.zip

https://doi.org/10.5281/zenodo.5033314

**User's Guide Documentation:**

The CARS version user's guide documentation can be accessed through the Github repository:the CARS version 1 used in this paper (Baek et al., 2021):

https://github.com/bokhaeng/CARS/tree/master/docs/User_Manual

https://doi.org/10.5281/zenodo.5033314

Lines 781-785

**References**

Baek, B. H., and Seppanen, C., SMOKE v4.8.1 Public Release (January 29, 2021) (Version SMOKEv481_Jan2021): http://doi.org/10.5281/zenodo.4480334 last 2021.Pedruzzi, Rizzieri, Wang, Chi-Tsan, Woo, Jung-Hun (2021). bokhaeng/CARS: CARS (Comprehensive Automobile Emissions Research Simulator) version 1.0 Public Release (CARSv1.0). Zenodo. https://doi.org/10.5281/zenodo.5033314

Baek, B. H., and Seppanen, C., SMOKE v4.8.1 Public Release (January 29, 2021). (Version SMOKEv481_Jan2021): http://doi.org/10.5281/zenodo.4480334.